

# Comparison of Emissions Inventories of Anthropogenic Air Pollutants in China

Eri Saikawa[1,2], Hankyul Kim[2], Min Zhong[1], Yu Zhao[3], Greet Janssens-Maenhout[4], Jun-ichi Kurokawa[5], Zbigniew Klimont[6], Fabian Wagner[7], Vaishali Naik[8], Larry W. Horowitz[8], and Qiang Zhang[9]

[1]Department of Environmental Sciences, Emory University, Atlanta, GA
[2]Rollins School of Public Health, Emory University, Atlanta, GA
[3]School of the Environment, Nanjing University, Nanjing, China
[4]European Commission, Joint Research Centre, Directorate of Energy, Transport and Climate, Via Fermi, 2749, 21027 Ispra (VA), Italy
[5]Asia Center for Air Pollution Research, 1182 Sowa, Nishi-ku, Niigata, Niigata, 950-2144, Japan
[6]International Institute for Applied Systems Analysis, Laxenburg, Austria
[7]Andlinger Center for Energy and the Environment, Princeton University, Princeton, NJ
[8]NOAA Geophysical Fluid Dynamics Laboratory, Princeton, NJ, USA
[9]Center for Earth System Science, Tsinghua University, Beijing, China

*Correspondence to:* Eri Saikawa (eri.saikawa@emory.edu)

**Abstract.** Anthropogenic air pollutant emissions have been increasing rapidly in China. Modelers use emissions inventories to assess temporal and spatial distribution of these emissions to estimate their impacts on regional and global air quality. However, large uncertainties exist in emissions estimates and assessing discrepancies in these inventories is essential for better understanding of the

trends in air pollution over China. We compare five different emissions inventories estimating emissions of carbon monoxide (CO), nitrogen oxides ($NO_x$), sulfur dioxide ($SO_2$), particulate matter with an aerodynamic diameter of 10 $\mu$m or less ($PM_{10}$) from China. The emissions inventories analyzed in this paper include Regional Emissions inventory in ASia v2.1 (REAS); Multi-resolution Emission Inventory for China (MEIC); Emission Database for Global Atmospheric Research v4.2 (EDGAR);

the inventory by Yu Zhao (ZHAO); and the Greenhouse Gas and Air Pollution Interactions and Synergies (GAINS). We focus on the period between 2000 and 2008 during which the Chinese economic activities have more than doubled. In addition to the national total, we also analyzed emissions from four source sectors (industry, transportation, power, and residential) and within seven regions in China (East, North, Northeast, Central, Southwest, Northwest, and South) and found that large

disagreements ($\sim$ seven fold) exist among the five inventories at disaggregated levels. These discrepancies lead to differences of 67$\mu$g/m$^3$, 15ppbv, and 470ppbv for monthly mean $PM_{10}$, $O_3$, and CO, respectively, in modelled regional concentrations in China. We also find that MEIC inventory emissions estimates create a VOC-limited environment that produces much lower $O_3$ mixing ratio in the East and Central China compared to the simulations using REAS and EDGAR estimates. Our

results illustrate that a better understanding of Chinese emissions at more disaggregated levels is



essential for finding an effective mitigation measure for reducing national and regional air pollution in China.

## 1 Introduction

Obtaining accurate emissions estimates for air pollutant species is important in Asia, where five of
the worst air quality countries in the world belong (Hsu et al., 2014). Emissions of ozone precursors, including nitrogen oxides ($NO_x \equiv NO + NO_2$) and carbon monoxide (CO), affect tropospheric ozone ($O_3$) mixing ratio at local, regional, and inter-continental scales (Fiore et al., 2009; West et al., 2009). In addition to the emissions of primary particulate matter (PM), those of sulfur dioxide ($SO_2$) and $NO_x$ also affect PM concentrations on local and regional scales. Both surface $O_3$ and PM are linked
to adverse health impacts (Dockery et al., 1993; Levy et al., 2001; Pope III et al., 2002), and $O_3$ also affects agricultural crop yields (Heck et al., 1983; Krupa and Manning, 1988; Avnery et al., 2011).

One key country in need of accurate emissions estimates is China, the largest emitter and the biggest contributor to the uncertainty in the source and the magnitude of many of the air pollutant species, such as $SO_2$ (Smith et al., 2011; Klimont et al., 2013). Energy consumption has been
steadily increasing in China but at the same time, the implementation of emissions control measures, including the flue-gas desulphurization (FGD) in coal-fired power plants, has led to rapid changes in emission factors in recent decades (Xu, 2011; Zhang et al., 2012; Kurokawa et al., 2013). Several emissions inventories have been developed in the past, either specifically for China or for Asia (Streets and Waldhoff, 2000; Streets et al., 2003; Zhao et al., 2008; Klimont et al., 2009; Lu
et al., 2010; European Commission, Joint Research Centre (JRC)/Netherlands Environmental Assessment Agency (PBL), 2011; Lei et al., 2011; Lu et al., 2011; Smith et al., 2011; Zhao et al., 2012; Kurokawa et al., 2013; Klimont et al., 2013) but none have assessed or compared emissions from different source sectors at more disaggregated scales than the national level.

The purpose of this study is to analyze the discrepancies among the existing emissions inventories for China's anthropogenic gaseous and aerosol emissions, including carbon monoxide (CO), sulfur dioxide ($SO_2$), nitrogen oxides ($NO_x$), and particulate matter with an aerodynamic diameter less than 10 $\mu$m ($PM_{10}$). We first evaluate the differences among inventories at the national level between years 2000 and 2008. Next, we focus on four source sectors (industry, transport, power, and residential) in seven regions of China (the East, North, Northeast, Central, Southwest, North-
west and South). By disaggregating emissions into these source sectors and regions, we aim to understand where the discrepancies occur and how we can better constrain emissions. We also use a chemical transport model, the Weather Research and Forecasting model coupled with Chemistry (WRF-Chem), to assess how these emissions discrepancies affect air quality modeling results.

The paper is organized as follows. Section 2 explains the emissions inventories that we have
compared. Section 3 analyzes the differences in emissions inventories first at the national level and



then in seven regions within China. Section 4 compares transport sector emissions in depth for CO and NO$_x$. Section 5 describes the impact of the emissions inventories on air quality simulations. Section 6 presents a summary of results and suggested future research.

## 2 Emissions Inventories

In this study, we compare five existing emissions inventories at the national, regional, and source sector levels between years 2000 and 2008 (Table 1). The Regional Emission inventory in ASia version 2.1 (REAS) is a regional emissions inventory for most of the Asian countries including the East, Southeast, South, and Central Asia and the Asian part of Russia (Kurokawa et al., 2013). The Emission Database for Global Atmospheric Research version 4.2 (EDGAR) is a global emissions

inventory and includes major air pollutants from combustion and non-combustion sources (European Commission, Joint Research Centre (JRC)/Netherlands Environmental Assessment Agency (PBL), 2011). Multi-resolution Emission Inventory for China (MEIC, http://meicmodel.org/) is an inventory developed at Tsinghua University, Beijing, China, and provides national emissions estimates for 2008 and 2010. A national emissions inventory for the 2007-2014 period was developed at Nan-

jing University (Zhao et al., 2008). The Greenhouse Gas and Air Pollution Interactions and Synergies (GAINS, http://gains.iiasa.ac.at/models/index.html) model is a framework for analyzing co-benefits reduction strategies from air pollution and greenhouse gas sources globally and emissions are calculated within the model (Amann et al., 2011). These five emissions inventories were developed using a similar methodology, where emissions were calculated as the product of activity data, such as fuel

consumption or industrial production, emission factors of combustion or production technology, and penetration rate and emission reduction efficiency of emission controls (Zhao et al., 2014). Table 2 shows how emissions in each of the inventories are aggregated to the four primary source sectors (industry, transport, power, and residential) that we analyze in this paper. They were grouped in this way to be able to compare at the four source sector levels among the inventories, as this is how some

of the inventories (i.e., MEIC) are structured. Here we explain each of the emissions inventories in more detail.

### 2.1 REAS

REAS was developed by collaboration between the National Institute for Environmental Studies and Asia Center for Air Pollution Research, Japan (Kurokawa et al., 2013). The inventory comprises of

emissions data from 30 Asian countries and regions, including China divided into 33 sub-regions, between years 2000 and 2008 at a 0.25° longitude x 0.25° latitude resolution. Previous version of REAS spanned a longer time period in the past and included projections of emissions (Ohara et al., 2007) but v2.1 is based on updated activity data and parameters. The emissions sources provided are





power plants, combustible and non-combustible sources in industry, on-road and off-road sources in
transportation, and residential and others such as agricultural activities and evaporative sources.

## 2.2 EDGAR

EDGAR was developed by the Joint Research Centre of the European Commission, in collaboration
with the Netherlands Environmental Assessment Agency (European Commission, Joint Research
Centre (JRC)/Netherlands Environmental Assessment Agency (PBL), 2011). This database incor-
porated experiences of the dataset EDGAR v3.2 FT2000 from Olivier et al. (2001). EDGAR is a
gridded emissions inventory of greenhouse gases, air pollutants and aerosols that spans 1970 - 2008
at a 0.1° longitude x 0.1° latitude horizontal resolution. Gridding of national total emissions is done
using several types of proxy data (population, road, power plants, animals, crop) as described in
Janssens-Maenhout et al. (2013). The source sectors provided are energy, industrial processes, prod-
uct use, agriculture, waste, and other anthropogenic sources. Country emissions are compiled based
on the International Energy Agency (IEA) energy statistics and Food and Agriculture Organization
(FAO) of the United Nations agriculture statistics. Emission factors are taken from the EMEP/EEA
air pollutant emission inventory guidebook (European Environment Agency, 2013) and other scien-
tific literature.

## 2.3 MEIC

MEIC is an inventory developed at Tsinghua University, Beijing, China, and provides source sector
information for each Chinese province for 2008 and 2010 (Li et al., 2014; Zheng et al., 2014; Li
et al., 2015; Liu et al., 2015). The MEIC model has fine spatial and sectoral resolution and allows for
gridding of the emission product into user-specific grid including 0.25° longitude x 0.25° latitude
horizontal resolution, as well as coarser grids. The emissions sources provided are power plants,
industry, transportation, residential and agricultural sources.

## 2.4 ZHAO

The inventory made at Nanjing University is a national inventory that estimates source sector emis-
sions from 31 Chinese provinces (Zhao et al., 2013b, 2015; Cui et al., 2015; Xia et al., 2016). The
inventory includes data for 2000-2014 but we use the disaggregated emissions estimates for 2007
for comparison. The sectors provided are industry (including cement, iron & steel, other industrial
combustion, and other industrial processes), power, transportation (including on-road and off-road),
and residential.

## 2.5 GAINS

The GAINS model was developed at the International Institute for Applied Systems Analysis and
estimates global emissions, including those for 31 provinces in China (Amann et al., 2008; Klimont



et al., 2009). The GAINS model calculates emissions estimates in five year intervals from 1990 to 2050, with the projection starting in year 2015. It has a large number of source sectors including energy, domestic, industrial combustion and processes, road and non-road transportation, and agri-

culture, for which activities originate from international and national statistics. It provides output in various formats and spatial resolution, including 0.5 $^\circ$ latitude x 0.5 $^\circ$ longitude horizontal grid. For this study, we use estimates from energy, domestic, transportation, and industry sectors for the years 2000 and 2005, using the global dataset developed within the European Union project ECLIPSE (version V5a, http://www.iiasa.ac.at/web/home/research/researchPrograms/air/Global_emissions.html)

(Klimont et al., in review; Klimont et al., in preparation).

### 3   National and Regional Comparisons

To better understand the discrepancies among anthropogenic emissions estimates of four air pollutant species, we first analyzed differences in national total emissions estimates between years 2000 and 2008. For each of the species, we further compared these estimates in seven different regions (Fig. 1)

for four source sectors separately. In the following sections, we first describe the discrepancies at the national level, and then at the regional level for each species.

### 3.1   National Total Comparisons

Fig. 2 illustrates China's national total emissions for the four air pollutant species of our interest (CO, $SO_2$, $NO_x$, and $PM_{10}$) as well as $CO_2$ estimated by REAS, EDGAR, MEIC, ZHAO, and GAINS,

between 2000 and 2008. We find the largest discrepancy, ranging 87 - 106%, between REAS and EDGAR emissions estimates for total CO in China throughout the 2000 - 2008 time period. The GAINS national total CO estimates lie almost in between those of EDGAR and REAS but the MEIC and ZHAO national emissions estimates are closer to the REAS estimate. Other published CO national emissions estimates are also close to REAS estimates. For example, Streets et al. (2003) esti-

mated 116 Tg/year for the year 2000, and Streets et al. (2006) estimated 151 Tg/year for 2001. Zhao et al. (2012) estimated 173, 179, 179, and 167 Tg/year for the years 2005, 2006, 2007, and 2008, respectively, and Zhang et al. (2009) estimated 167 Tg/year for 2006. These are all well-aligned with the REAS estimates. Top-down estimates, optimizing the emissions using both observational data and the simulations from atmospheric chemical transport models, for the early 2000s were also

high, ranging between 140 and 230 Tg/year (Streets et al., 2006; Tanimoto et al., 2008). EDGAR underestimates CO emissions, especially in industry and transportation sectors because the modeling of superemitters have been omitted and this seems more important for emerging countries. The discrepancies are apparent in Fig. 3.

The discrepancy for $PM_{10}$ between REAS and EDGAR is also not insignificant and ranges be-

tween 25 and 59% over time. Similar to CO, the national estimates (MEIC and ZHAO) and GAINS



are all closer to the REAS estimate. For example, Zhang et al. (2009) estimated 18.2 Tg/year in 2006, which is close to the 20.0 Tg/year estimate in REAS for the same year, compared to the 12.7 Tg/year estimate in EDGAR. The REAS estimate is also comparable to the 18.4 Tg/year estimated in ZHAO for 2007. Some estimates for the earlier years are higher than those of REAS. Zhang et al. (2009)

estimated 16.1 Tg/year for 2001, larger than the REAS estimate of 14.2 Tg/year for the same year. As shown in Fig. 3, the discrepancies arise mainly from the industry sector, where EDGAR emissions show significantly lower estimates compared to those of REAS and by Zhang et al. (2009). Power sector emissions show the opposite trend and EDGAR emissions are double those of REAS and Zhang et al. (2009), which is most likely due to the lack of consideration of emissions reduction

technologies in EDGAR as mentioned later in more detail.

The discrepancies for the other species are much lower. The range of the difference between REAS and EDGAR for $CO_2$, $SO_2$, and $NO_x$ are 0.07 - 7%, 2.1 - 20%, and 7.3 - 27%, respectively, between 2000 and 2008. MEIC and ZHAO emissions estimates fall between the REAS and EDGAR estimates most of the time, although they are again closer to the REAS estimates, which are higher

than those of EDGAR, for most species. GAINS estimates sometimes do not fall between the REAS and EDGAR estimates but the discrepancies are still low. The timing of the $SO_2$ emissions reduction in 2007 in REAS coincides with what is reported in Zhang et al. (2009), Klimont et al. (2009), and Lu et al. (2011). Smith et al. (2011) estimated 2000 and 2005 national $SO_2$ emissions to be 21.4 and 32.7 Tg/year, close to the REAS (GAINS) estimates of 22.2 (24.2) Tg/year and 34.1 (32.4) Tg/year,

respectively. The EDGAR $SO_2$ estimate of 19.8 Tg/year in 2000, however, is closer to the official estimate of 19.95 Tg/year by State Environmental Protection Administration (SEPA) (2000) and the estimate by the Streets et al. (2003) of 20.4 Tg/year, compared to the REAS estimate of 22.2 Tg/year.

### 3.2 Regional Total Comparisons

When we compare emissions in the seven regions within China (East, North, Northeast, Central,

Southwest, Northwest, and South, as shown in Fig. 1), we find larger differences than at the national level for almost all species. We compare in detail the differences among emissions inventories for each species per region and for each source sector below.

#### 3.2.1 Carbon monoxide, CO

Atmospheric CO is mainly a result of incomplete combustion of fossil fuels and biofuels. Expo-

sure to ambient CO is harmful to human health (Aronow and Isbell, 1973; Stern et al., 1988; Allred et al., 1989; Morris et al., 1995) and CO emissions are also important precursors to the formation of tropospheric $O_3$, which also has harmful human health impacts (Schwartz et al., 1994). Because of the existence of diverse emissions sources with various emissions control technologies in China, it has been a challenge to estimate CO emissions accurately, using a bottom-up methodology with

emission factors and activity levels (Streets et al., 2006). This explains why we see the largest dis-





crepancy in CO emissions estimates at the national level as we found in Fig. 2 amongst all other species.

Fig. 4 shows the national and seven regional CO emissions estimates from each source sector. For CO emissions, industry is the only source sector that shows a steep increase over time in all regions for REAS and EDGAR estimates, especially between 2002 and 2008. GAINS also shows an increase between 2000 and 2005. For the national total emissions, we find a 105% (132%) increase for REAS (EDGAR) estimates in 2008 from 2000 values. Due to this rapid increase, industry is the largest source sector for CO in the two largest source regions - East and North - by 2008, regardless of the inventories. Industry emissions share 52, 53, and 42% of the national total in REAS, EDGAR, and MEIC, respectively. Similarly, they share 48% (43%) of the national total in ZHAO (GAINS) estimates in 2007 (2005). REAS CO emissions estimates are consistently higher than those of EDGAR across all regions except for the Northeast for industry emissions, and MEIC, ZHAO, and GAINS CO emissions estimates for this sector generally fall between the estimates of REAS and EDGAR. The two regions where this does not apply are Central and Northwest, and their industrial CO emissions estimates by MEIC, ZHAO, and GAINS are higher than the estimates by the other two emissions inventories. Analysis at the source sector level reveals that the majority of the differences in CO emissions among the inventories stem from the industry sector and that they are, in many regions, increasing over time. Indeed, between REAS and EDGAR, 38% of the difference in national total CO emissions stems from the industry sector in 2000. By 2008, the industry sector shares 51% of the difference in their estimates.

What brings such a large discrepancy from the industry sector? Coal combustion plays a large role in CO emissions from this sector in the REAS estimate and indeed, 98.6% of the combustible industrial emissions are due to coal in 2008. It is possible that both emission factors and activity data have large uncertainties in industrial CO emissions, but considering it is not obvious in $SO_2$ or $NO_x$ emissions but only in CO and $PM_{10}$, as mentioned later, we find that the discrepancies are not due to activity level or fuel use, which should affect all emissions. Instead, these discrepancies are due to assumptions of production technology and combustion efficiency for CO. Because emission factor is related to each technology type, penetration of the technology, uncontrolled emission factor and the emission reduction efficiency of each technology type, all contribute to discrepancies. Obtaining estimates for CO is particularly troublesome because of many technology types that exist for emissions reduction. Furthermore, REAS uses averaged emission factors in ton/kt in four categories for coal and five categories for industrial processes, whereas EDGAR, MEIC, and ZHAO all use disaggregated emission factors for each emission standard level and calculate emissions per energy consumed (g/TJ in EDGAR) or per fuel burned (g/ton in MEIC and ZHAO). Inventories using different units for emission factor like this adds uncertainty with respect to assumptions about the fuel heat value, which has been changing over time.



The second largest CO source is the residential sector and the estimates by the national inventories MEIC and ZHAO are always higher in all regions than the regional inventory REAS and the global inventory EDGAR estimates. GAINS estimates the residential sector to be the largest source sector

and these emissions share 64 and 52% of the national emissions in 2000 and 2005, respectively. Their estimates are also usually higher than REAS and EDGAR in almost all regions, except in the Southwest and the South in 2005, where the REAS and GAINS estimate are close to each other. EDGAR estimate for residential sector emission is the lowest among the inventories analyzed here because it does not include provincial but rather uses the national statistics-based IEA estimates for coal use in

residential sector, leading to lower activity level. On the other hand, GAINS emissions for this sector are the highest because it is unique in considering factors which are technology specific, rather than one factor per whole residential sector and fuel. For example, there are significant differences in emissions for different types of stoves and boilers in residential sector and these technology-specific data are incorporated into the GAINS model.

The third largest CO source and the source sector with the second largest discrepancy after industry is transportation, sharing 43.7% of the difference in 2000 and 34.4% in 2008. Emissions from North and East regions contribute to these large discrepancies. Both REAS and EDGAR emissions inventories show a decreasing trend at the national level, although at the regional scale, the rate is -41% (-44%) for EDGAR and -20% (13%) for REAS in 2008 compared to 2000 in the North (East).

This discrepancy might be due to a couple of reasons. First, emission factors and reduction measures assumed can be different. For example, EDGAR may be estimating much larger emissions reduction in newer vehicles with more stringent emission standards. Second, the number of vehicles assumed in different vehicle types can be different among the inventories, even if the total number may be similar. For REAS, the number of vehicles of each type (passenger cars, buses, light and heavy duty

trucks, and motorcycles) in 2000 was taken from Borken et al. (2008) and extrapolated to 2008, using trends from National Bureau of Statistics (2001-2009) (Kurokawa et al., 2013). Emission factors due to control strategies and policies in REAS stem from estimates in Borken et al. (2008) and Wu et al. (2011), as explained in Saikawa et al. (2011). For EDGAR, the fleet distribution is based on the international statistics from the International Road Federation (IRF, 1990, 2005, 2007) which

were analyzed in the framework of the EU 'Quantify' project (Borken et al., 2008). Zhang et al. (2009) estimated 11% decrease in CO from the transportation sector between 2001 and 2006 due to emissions control technologies, despite the doubling of the number of vehicles in the same period. We will analyze the transportation emissions in more detail in Section 3.3 as we have some more disaggregated data for this sector available for comparison.

The smallest CO source sector is power. It only contributes to 1.9, 3.1, 1.1, and 0.8% of the national emissions in REAS, EDGAR, MEIC, and ZHAO, respectively, in 2008 for the former three and in 2007 for ZHAO. GAINS estimates 1.2% of its national emissions comes from power in 2005. REAS estimates 159% increase in CO emissions from the power sector between 2000 and



2008, while EDGAR only estimates 15% increase in the same time period. At the national level, the
discrepancy in CO emissions from the power sector between REAS and EDGAR decreased from
50% to 13% between the same period (2000-2008). At the regional level, CO emissions from the
power sector are small but the actual discrepancy is increasing for all regions over time, except in
the Northwest.

At the national level, CO emissions are ranked first by industrial, next by residential, then by
transportation, and power. At the regional level, however, this ranking of source sectors does not al-
ways hold. For Northwest, emissions from the residential sector are estimated to be larger than those
from industry in all inventories. In Southwest, REAS estimates higher industrial emissions than res-
idential emissions but EDGAR estimates higher transportation emissions than industrial. Similarly,
in the South, REAS estimates industry to be the largest source sector followed by residential and
transportation, whereas EDGAR estimates residential to be the largest, followed by industry as a
close second and transportation with much lower emissions than the other two in the recent years.
This clearly illustrates the importance of constraining emissions at the disaggregated levels.

The East, encompassing the Pearl-River-Delta and the industrial coast, is the largest source re-
gion of CO and it shares 32, 27, and 26% of the national total CO emissions from REAS, EDGAR,
and MEIC estimates, respectively, in 2008. Similarly, ZHAO (GAINS) estimates 30% (28%) of the
national total CO emissions is from the East in 2007 (2005). CO emissions from the industry sec-
tor in the East, in particular, show a high level of discrepancy, and the absolute difference more
than doubles from 2000 to 2008. 22.4 Tg/year CO emissions discrepancy in the industry sector in
2008 between REAS and EDGAR constitutes 64% of the difference between the two emissions es-
timates within the East in that year. This discrepancy makes up 25% of the difference between the
two national total CO emissions estimates. The difference between the REAS and EDGAR emis-
sions estimates for the transportation sector for this region is also increasing and is 10.1 Tg/year in
2008, equivalent to 29% of the regional total CO difference and 11% of the national CO difference.
One thing to note about this region is that EDGAR CO estimates for the transportation sector are
decreasing over time, whereas those of REAS indicate the opposite.

The North is the second largest source region of CO, and it shares 21, 14, and 21% of the national
total CO emissions for REAS, EDGAR, and MEIC estimates, respectively, in 2008. ZHAO (GAINS)
estimates 18.5% (18.1%) of the national total CO emissions from this region in 2007 (2005). Com-
bined with the East emissions, the two regions share 53, 42, 47, 48, and 46% of the emissions in
REAS, EDGAR, MEIC, ZHAO, and GAINS, respectively, in 2008 for the former three, 2007 for
ZHAO, and 2005 for GAINS. The pattern shown for East and North, the more developed regions
in China, is similar, and the only difference is that EDGAR estimates larger residential emissions
compared to transport emissions in the East, whereas the opposite is the case for the North in the
early 2000s.



### 3.2.2 Sulfur dioxide, SO$_2$

$SO_2$ leads to acid rain through sulfuric acid deposition, destroying buildings by corroding metals and deteriorating paint and stone. Furthermore, it harms aquatic and terrestrial ecosystems. $SO_2$ is also a precursor of sulfate aerosols that scatter radiation, leading to direct cooling of the atmosphere. Sulfate aerosols also act as condensation nuclei, making clouds more reflective and prolonging the lifetime of clouds, enhancing the cooling impact (Haywood and Boucher, 2000; Ramanathan et al., 2001).

Fig. 5 shows the national and seven regional $SO_2$ emissions estimates for each source sector. For $SO_2$ emissions, the power sector is the largest source sector in most years for both REAS and EDGAR, and 38 - 54% (52 - 61%) of national total $SO_2$ emissions are from the power sector in REAS (EDGAR) between 2000 and 2008. Contrary to CO emissions, we find a large divergence between REAS and EDGAR power sector emissions estimates during 2000 - 2008 across all regions. While EDGAR $SO_2$ power emissions estimates continue to increase over time, those of REAS peak in that time range, although the specific year is not uniform across the regions. The Central and the Northwest start to deviate in 2004, the South, East, and North in 2005, and the Northeast and the Southwest in 2006.

The large discrepancy in $SO_2$ emissions from the power sector between REAS and EDGAR is due to the difference in the assumed timing of the installation of FGD in coal-fired power plants. Newly designed policy incentives and an increase in policy inspection have led to an increase in the installation of FGD in China and the percentage of plants with FGD increased from 10% to 71% between 2006 and 2009 (Xu, 2011). The number of power plants is listed in Table 3. While EDGAR assumed a delayed penetration of FGD (1%), electrostatic precipitators (6%) and flue-gas recirculation (4%) leaving 90% of power plants still fully-uncontrolled in 2008, REAS estimated a more optimistic installation scenario, especially for large power plants and referred Lu et al. (2011) in deciding implementation rates of FGD to power plants in China. For example in 2007, Lu et al. (2011) used the range of 51.4 - 95%, with the mean of 73.2%, based on the Chinese Ministry of Environmental Protection (MEP) official data (2009) reporting of $SO_2$ removal efficiency of FGD and applying the triangular distribution with the ideal removal efficiency of 95% (Zhao et al., 2011). This explains why REAS emissions estimates from the power sector are closer to the national emissions estimates by MEIC, and those by Lu et al. (2011), as seen in Fig. 6. The largest emissions decrease from the power sector are seen in the East and North regions, where there were 250 and 206 power plants, respectively, reinforcing that this discrepancy is due to the FGD implementation assumption in power plants.

The second largest source sector for China's $SO_2$ emissions is industry. Nationally, it shares 53, 33, 53, 44, and 27% of total $SO_2$ emissions in REAS, EDGAR, MEIC for 2008, ZHAO for 2007, and GAINS for 2005, respectively. In some regions, there is very little discrepancy among inventories, for example, in the Northeast. On the other hand, we see a much larger difference in the Southwest.





While EDGAR estimates industry to be the second largest source sector in this region, constituting 31 - 37% of regional emissions, all other emissions inventories estimate industry to be the largest source sector in the region, constituting 46 - 60% of the regional total. Similar to its estimates for CO emissions, REAS tends to estimate higher emissions from the industry sector in most of the regions. In all regions other than the South, industrial $SO_2$ emissions estimates by MEIC, ZHAO, and GAINS lie between REAS and EDGAR estimates.

$SO_2$ emissions discrepancies in the two other sectors remain relatively constant across all regions, with the residential sector emissions in the Southwest as the only exception. The percentage of residential sector emissions difference in the Southwest between EDGAR and REAS estimates has decreased from 113% in 2000 to 62% in 2008.

### 3.2.3 Nitrogen oxides, $NO_x$

$NO_x$ plays an important role in the formation of tropospheric $O_3$ and nitrate aerosols. The $NO_x$ emissions trend in Asia, and especially in China, has been an important topic, due to the rapid changes that have been observed in the past two decades (Richter et al., 2005; Gu et al., 2014).

Fig. 7 shows the national and seven regional $NO_x$ emissions estimated for each source sector. The power emissions dominate the national total and the regional totals, for REAS, EDGAR, and GAINS. 41% (46%) and 45% (51%) of the national $NO_x$ emissions are estimated to come from the power sector in REAS and EDGAR, respectively, in 2000 (2008). 47% of the total emissions are estimated to come from the power sector in 2005 for GAINS. The national emissions inventories, however, do not show the increasing trend and for MEIC, industrial emissions are estimated to be higher than those from the power sector. For ZHAO, the two sources are similar in magnitude. 33% (36%) and 35% (35%) of the total emissions are estimated to come from the power (industry) sector in these two national inventories MEIC in 2008 and ZHAO in 2007, respectively. One of the reasons for this discrepancy is that there is a systematic difference between the national and IEA statistics in terms of how fuel use is reported in the power or energy sector. While EDGAR and GAINS use the IEA statistics, all others use the Chinese provincial statistics for fuel use. Combined with the difference in assumed activity levels and the emission factors, this difference in fuel use statistics is leading to the discrepancy that is not necessarily consistent in different regions. Streets et al. (2003) estimated power to be the dominant source sector, sharing 39% of $NO_x$ emissions in 2000, followed by equal 25% contribution from industry and transportation. The discrepancy in this sector is the largest in the East and the Northeast regions, both with 470 Gg/year in 2008. The fact that $NO_x$ emissions estimates from various inventories have similar trends and do not show discrepancies as in $SO_2$ further confirms that the discrepancy in $SO_2$ emissions from the power sector is due to FGD implementation and not to activity levels.

The large discrepancies among the emissions inventories stem from the transportation sector in the East, South, Northwest, and Southwest. For the transportation sector, the East has an increasing



discrepancy over time, changing from 41% in 2000 to 61% in 2008. While transportation shares 27 -
30% of the regional total emissions for REAS in the East, it only shares 15 - 19% for EDGAR. MEIC

estimates the transportation sector in the East to share 25% of the regional total $NO_x$ emissions. In
the South, Northwest, and Southwest, the discrepancy from this sector can also be as high as 67, 72,
and 83%, respectively. The key reasons why the differences are large and they are growing are two-
folds. First, this is because of the difference in allocation of fuel (gasoline and diesel) and to different
vehicle categories, as we explain later in Section 4. Second, it is because the pace of implementation

of measures assumed among different inventories is different.

Little to no emissions control technologies for $NO_x$ has been developed and promoted in China
for the power and industrial combustion sectors and this is the main reason why we see a large
upward trend for $NO_x$ emissions. China only used low $NO_x$ combustion technology and started to
install selective reduction methods after 2005 (Zhao et al., 2013a). The only other $NO_x$ mitigation

strategy for China was emissions standards for reducing tail pipe emissions from vehicles (Zhao
et al., 2013a). For example, there is no national $NO_x$ emissions standard for coal-fired industrial
boilers, as opposed to the vehicle emission standards that have been tightened over the years.

The industry and transport sectors are next after the power sector, sharing equal portions in both
REAS and EDGAR. REAS (EDGAR) estimates 23-27% (21-25%) from the transport sector and

23-30% (18-25%) from industry. Over time, the power sector is increasing its share, similarly for the
industry. A slight decrease is seen for both inventories in the transport sector. For $NO_x$, we find that
all the regions show similar trends.

### 3.2.4    Coarse particulate matter, $PM_{10}$

China's PM emissions have been increasing rapidly and they share approximately 65% (38%) of

emissions from 22 Asian countries, including Afghanistan, Bangladesh, Bhutan, Nepal, Sri Lanka,
India, Maldives, Pakistan, South Korea, North Korea, China, Japan, Singapore, Taiwan, Laos, Cam-
bodia, Brunei, Myanmar, Philippines, Thailand, Vietnam, and Indonesia, in the REAS (EDGAR)
estimate. Here, we only discuss primary emissions, PM emitted directly from anthropogenic sources.

Fig. 8 shows the national and seven regional $PM_{10}$ emissions estimates for each source sector. The

largest source sector, as well as the largest emissions discrepancy, stems from the industry sector.
Industrial emissions share 64, 19, and 78% of the total $PM_{10}$ emissions in REAS, EDGAR, and
MEIC for 2008, respectively, and 65% (51%) for ZHAO (GAINS) for 2007 (2005). Although the
industrial emissions share in 2008 in EDGAR is lower than that of the others, the EDGAR industry
share has gone up by 6 percentage points from 2000 to 2008, similar to REAS with 8 percentage

points increase in the same period. The reason for this large increase in industrial $PM_{10}$ emissions
is due to the fast growth of industry and limited stringency of air quality legislation and its poor
enforcement (Zhao et al., 2013a). In addition, uncertainty accounting for fugitive emissions due to
leaks or other unintentional releases adds to the discrepancy among the inventories. For industrial



PM$_{10}$ emissions, REAS estimates are always consistently higher than those of EDGAR in all regions,
and the difference between the two inventories is four to five-fold, constituting 61 - 74% of the total
differences.

The large discrepancy in industrial PM$_{10}$ emissions is due to differences in both emission factors
and emissions reduction factors embedded in emission calculations among inventories. We find a
large emissions discrepancy in CO and the trend holds well for PM$_{10}$ as well. At the same time,
similar to CO, 95 - 96% of industrial PM$_{10}$ emissions in REAS come from hard coal and it is most
logical that the five emissions inventories estimated different emission factors and emission reduc-
tion factors for industrial emissions. Obtaining estimates for PM$_{10}$, just as for CO, is particularly
troublesome compared to SO$_2$ or NO$_2$, which have limited technology types for emissions reduc-
tion.

The second largest discrepancy comes from the power sector. Power emissions share 13, 45, and
12% of the total PM$_{10}$ emissions in REAS, EDGAR, and MEIC in 2008, respectively, and 9%
(14%) for ZHAO (GAINS) in 2007 (2005). We find the largest discrepancy in the East among the
five inventories. It is also important to point out that the spatial distribution of emissions in some of
the inventories, especially the international ones, are often more static than the national ones due to
the limited local information, although this also applies to other species as well.

We see relatively little change in differences among the inventories between 2000 and 2008 for
transportation and residential sectors. There are, however, some interesting sector-dependent differ-
ences. First, GAINS estimates higher residential emissions than REAS and EDGAR in all regions
in both 2000 and 2005 except in the South in 2005. Second, REAS estimates are not always higher
than those of EDGAR for the residential sector emissions. In the Northeast, REAS PM$_{10}$ emissions
estimates are higher than those of EDGAR. For the Southwest and the North, REAS emissions esti-
mates are higher than EDGAR estimates only for the period 2002 - 2005. What is also striking is the
very small magnitude of residential sector PM$_{10}$ emissions estimated in MEIC, compared to other
inventories.

## 4 Transportation Sector Comparison

Rapid growth in the number of vehicles has created a significant air quality challenge in China. Many
have researched the importance of on-road transportation emissions on Beijing's (Hao et al., 2001;
Westerdahl et al., 2009) and China's air quality (Fu et al., 2001; Walsh, 2007; Saikawa et al., 2011).

Fig. 9 shows the on-road transport emissions from different vehicle types for CO, NO$_x$, PM$_{10}$, and
SO$_2$ estimated by REAS and EDGAR in 2008 for China. It is clear that the majority of emissions
(62%) come from gasoline cars and motorcycles in REAS and almost all in EDGAR for CO, whereas
almost all emissions (77% and 98%) come from other diesel vehicles for NO$_x$ in REAS and EDGAR,
respectively. For PM$_{10}$, rest of gasoline vehicles (e.g., buses) shares almost all emissions in REAS




and the majority (80%) in EDGAR. EDGAR estimates significant $SO_2$ emissions (60 Gg/year) from
445 transportation but they are non-existent in REAS.

As stated earlier for the industrial sector, it is likely that emission factors and/or the technology
levels estimated within each of the vehicle type are causing discrepancies. The lack of modeling
superemitters in EDGAR is also contributing significantly to the discrepancies. It is also important
to keep in mind that something more fundamental, such as the definition of vehicle types, is possibly
causing the discrepancies.

We found significant discrepancies in CO, $PM_{10}$, and $SO_2$ emissions in the transportation sector
and we analyzed the differences for CO and $NO_x$ emissions in more depth because we were able to
disaggregate these to on-road and off-road emissions. Here, we explain the discrepancies we find for
each of the species.

### 455   4.1   Carbon monoxide, CO

Fig. 10 shows the national and seven regional CO transportation emissions estimated in REAS,
EDGAR, and ZHAO, separated into on-road and off-road emissions and it shows clearly that the
discrepancy in this sector stems from on-road emissions. 99% of the difference between REAS and
EDGAR CO transportation emissions are from on-road at the national level, and in the East, we see
up to a difference of 99.4% at the regional level. Indeed, at the national and all regional levels, there
is more than an order of magnitude of difference in emissions between REAS and EDGAR on-road
emissions. ZHAO on-road emissions estimates are always in between REAS and EDGAR estimates
and ZHAO off-road estimates are always higher than both REAS and EDGAR.

### 4.2   Nitrogen Oxides, $NO_x$

Fig. 11 shows the national and seven regional $NO_x$ transportation emissions estimated in REAS,
EDGAR, and ZHAO, separated into on-road and off-road emissions. Contrary to the CO emis-
sions, there are many regional differences in these emissions estimates. At the national level, REAS
(ZHAO) estimates 42-56% (49%) higher for on-road emissions compared to EDGAR. Off-road
emissions are much more constrained among the three emissions inventories and REAS and EDGAR
give similar estimates between 2005 and 2007.

The East is estimated to share 28-38, 6.3-6.8, and 37% of the total transportation emissions in
REAS, EDGAR, and ZHAO, respectively. REAS (ZHAO) emissions estimates are 5.6-7.4 (6.2)
times larger than EDGAR on-road emissions, and 2.6-9.5 (6.7) times larger than off-road emissions.
For $NO_x$ emissions, although on-road emissions are still larger in most of the regions, off-road emis-
475 sions are also important and are mostly increasing in both REAS and EDGAR. For the East, REAS
estimates an increase of 258% in off-road emissions between 2000 and 2008. For the Northwest,
EDGAR estimates larger emissions from off-road compared to on-road for $NO_x$, which we do not
see in either REAS or ZHAO. REAS estimates a higher growth rate for off-road emissions and



their emissions estimates increase by 217% from 2000 to 2008, while EDGAR off-road emissions
estimates only increase by 16% over the same time period.

## 5 Impacts on air quality

### 5.1 Model description

To assess how these differences in emissions inputs affect air quality simulation results, we used the
Weather Research and Forecasting model coupled with Chemistry (WRF-Chem) version 3.5 (Grell
et al., 2005). The model domain covers much of the Asian region, with a horizontal resolution of 20
$\times$ 20 km and 31 vertical levels and China at its center (Fig. 12a). The initial and lateral chemical
boundary conditions are taken from a present-day simulation of the NOAA Geophysical Fluid Dy-
namics Laboratory (GFDL) global chemistry-climate model AM3 (Naik et al., 2013), driven by the
global gridded emissions from the inventory of Lamarque et al. (2010). The meteorological data are
obtained from the National Center for Environmental Prediction (NCEP) Global Forecast System
final gridded analysis datasets. We used Carbon-Bond Mechanism version Z (CBMZ) (Zaveri and
Peters, 1999) for gas-phase chemistry and the Model for Simulating Aerosol Interactions and Chem-
istry (MOSAIC) (Zaveri et al., 2008) for aerosol chemistry. The rest of the model setup (aerosol dry
deposition, wet deposition, photolysis, radiation, and microphysics) is the same as applied in our
previous study (Zhong et al., 2015).

We chose the three emissions inventories (REAS, EDGAR, and MEIC) for anthropogenic sources
of gaseous pollutants and PM and performed model simulations for January and July for 2008.
Because MEIC only covers China, we applied REAS emissions outside of China for the simula-
tion with MEIC. For biomass burning emissions, we used the Fire INventory from NCAR (FINN)
(Wiedinmyer et al., 2011) and for biogenic emissions, we used the Model of Emissions of Gases
and Aerosols from Nature (MEGAN) interactively within WRF-Chem (Guenther et al., 2012). For
aircraft emissions, we used emissions developed for the Hemispheric Transport of Air Pollution
(HTAP) for the year 2008 (Janssens-Maenhout et al., 2015). Dust emissions are not included in our
simulations but sea salt is calculated online (Gong, 2003). Before the beginning of each monthly
simulation, the model was spun-up for ten days to ventilate the regional domain.

### 5.2 Simulated results and discussion

Fig. 12a illustrates the spatial distribution of January emissions for CO, $NO_x$, $SO_2$, $PM_{10}$, and non-
methane VOC (NMVOC) that we used as inputs for the WRF-Chem simulations. As mentioned
earlier, CO and $PM_{10}$ show high variations and the emissions are especially concentrated in the
eastern part of China. Although $SO_2$ national discrepancy was not as large as those of the other
two species, Fig. 12a clearly illustrates that REAS estimates much larger emissions compared to the
other two inventories.





Fig. 13a compares the simulated monthly mean $PM_{10}$ concentrations, as well as that of CO, $NO_2$, $SO_2$, and $O_3$ mixing ratios in January 2008, using the three inventory estimates as emissions inputs.

Overall, the simulated monthly means show similar spatial distributions. All three simulations show high levels of CO, $NO_2$, $SO_2$, and $PM_{10}$ in the Beijing-Tianjin-Hebei area in the North, Shanxi province in the North, and Sichuan basin in the Southwest. In contrast, the mixing ratios of $O_3$ are relatively low over the same regions. Despite the similar spatial distributions, magnitudes of the simulated monthly means differ substantially.

For CO, both simulations using REAS and MEIC result in higher mixing ratios than when using EDGAR, showing differences of 65 - 122% in the polluted area in the Central and 81 - 89% in the East. 470 ppbv difference was found in Central China between simulations using EDGAR and MEIC. For $NO_2$, the largest differences occur between simulations using EDGAR and MEIC emissions, mainly in the Central (116%), followed by the Northeast (96%) and the East (91%). These

regions are where the differences in emissions are the largest as well. For $SO_2$, both simulations using REAS and MEIC show differences less than 30% in most regions compared to those with EDGAR emissions, except in the Southwest, where REAS and MEIC estimates are 43 and 50% higher, respectively, than EDGAR estimates.

For $PM_{10}$, EDGAR simulation is 20 - 60 $\mu g\,m^{-3}$ lower than the other two in most regions. For

example, MEIC simulation estimates 15 $\mu g\,m^{-3}$ (103%) higher in the Northeast and 20 $\mu g\,m^{-3}$ (85%) higher in the Southwest than EDGAR. REAS simulation estimates more than 55% higher $PM_{10}$ concentrations than EDGAR in most regions, with the highest difference (76%) occurring in the Northeast. The largest absolute difference of 67 $\mu g\,m^{-3}$ between MEIC and EDGAR simulations is found in the Central region. Based on the observations from nine stations in Wuhan within the

Central region, the monthly mean $PM_{10}$ concentrations in January were 130 $\mu g\,m^{-3}$ (Feng et al., 2011) and this is closer to the simulated values using the MEIC (REAS) emissions inventory of 47.4 (50.6) $\mu g\,m^{-3}$, compared to the value using the EDGAR emissions inventory of 32.3 $\mu g\,m^{-3}$, although the model simulations are largely underestimated.

For $O_3$, simulations using REAS and EDGAR inputs show only a slight difference of 1-5 ppbv in

January. However, $O_3$ mixing ratios using MEIC emissions are much lower than those using EDGAR emissions in the Central (31%) and the East (25%). MEIC's low VOC emissions in combination with high $NO_x$ emissions in these regions (see 12a) bring much higher $NO_x$ titration and produce a VOC-limited environment. It is well-illustrated in Figure 14a. For these two regions, despite the REAS and MEIC having similar $NO_x$ emissions, their VOC emissions differ by more than 10 times.

EDGAR emissions are the lowest for $NO_x$ for both the Central and the East but their estimates are the largest for VOCs in the Central and the second largest in the East among the three inventories. In both cases, simulations using EDGAR inventory lead to the largest $O_3$ mixing ratios, due to the limited titration of $NO_x$ during the night time. $NO_x$ mixing ratio in these two regions estimated in EDGAR is much lower compared to that in REAS and MEIC, as seen in Figure 13a. This result



illustrates the importance of VOC emissions estimates, in addition to $NO_x$ and other species that we have analyzed in this paper. Constraining these in the two regions will be essential in understanding the way to mitigate $O_3$ pollution for the future.

    We also analyzed the differences of three simulations in July 2008 (Fig. 13b). We find difference of more than 50% for CO, $NO_2$, $SO_2$, and $PM_{10}$ in one or more regions, while a difference is less

than 20% for $O_3$ in every region. The Central and the East again showed the largest differences, as found in January. There was 34 $\mu g\ m^{-3}$ difference in $PM_{10}$ in Central China between REAS and EDGAR and 129 ppbv difference in the East for CO between REAS and MEIC. Again, Wuhan mean for July of 70 $\mu g\ m^{-3}$ was better captured by MEIC (REAS) of 52.0 (53.5) $\mu g\ m^{-3}$, compared to that by EDGAR of 36.0 $\mu g\ m^{-3}$.

More detailed comparisons are illustrated in Table 4. These differences in simulated concentrations or mixing ratios of pollutants are solely due to the emissions used as model inputs. Not surprisingly, the results demonstrate that the choice of emissions inventories has a large influence on air quality simulation results and reinforce the need for better constraints on emissions inputs.

## 6   Conclusions

In this study, we compared five emissions inventories of anthropogenic $CO_2$ and air pollutant emissions in China at national and regional levels from four source sectors. The REAS and EDGAR inventories have been developed and maintained for years and have been extensively used for air quality modeling over the Asian continent, while the two of the national emissions inventories (MEIC and ZHAO) were recently developed and not many air quality modeling studies have been published

using the data at this time. GAINS has its roots in the RAINS-Asia model dating back to early 90's project covering primarily $SO_2$ and later on developed to include more pollutants. The dataset used here originates from a global project and has been used in several air quality and climate modeling exercises. This analysis reveals large discrepancies in emissions estimates among the existing inventories. Analysis of regional and sector specific emissions, as opposed to total national emissions,

reveals discrepancies in emissions from certain sectors that would not have been noticed by only analyzing the national total emissions.

    We find that there is an important need to better constrain emissions at the source sector and regional levels. Transparency in what inputs are used to create different emissions inventories becomes critical for a more thorough comparison. CO emissions differ the most and those from transport sec-

tor, especially the on-road transport emissions, need to be better constrained. Industrial emissions also tend to have a large discrepancy among inventories and $SO_2$ emissions from the power sector also need to be assessed, especially for recent years. The East and the North are the two largest emitting regions and more efforts are needed to understand emissions from these areas.





Emissions inputs have a large impact on air quality simulation results in China nationally, and
more prominently within the regions. Different emissions inputs lead to 67 $\mu g\ m^{-3}$ (34 $\mu g\ m^{-3}$)
monthly mean difference in $PM_{10}$ concentrations in Central China in January (July). Similarly, we
found 470 ppbv difference in January in Central and 129 ppbv difference in July in the East for
CO. We also found that MEIC inventory emissions estimates create a VOC-limited environment
in the Central and the East that produces much lower $O_3$ mixing ratio estimates, compared to the
simulations using REAS and EDGAR estimates. The discrepancy leads to 15 ppbv difference in
$O_3$ in Central China. Our results illustrate that a better understanding of Chinese emissions at more
disaggregated levels is essential for finding an effective mitigation measure for reducing national and
regional air pollution in China.



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



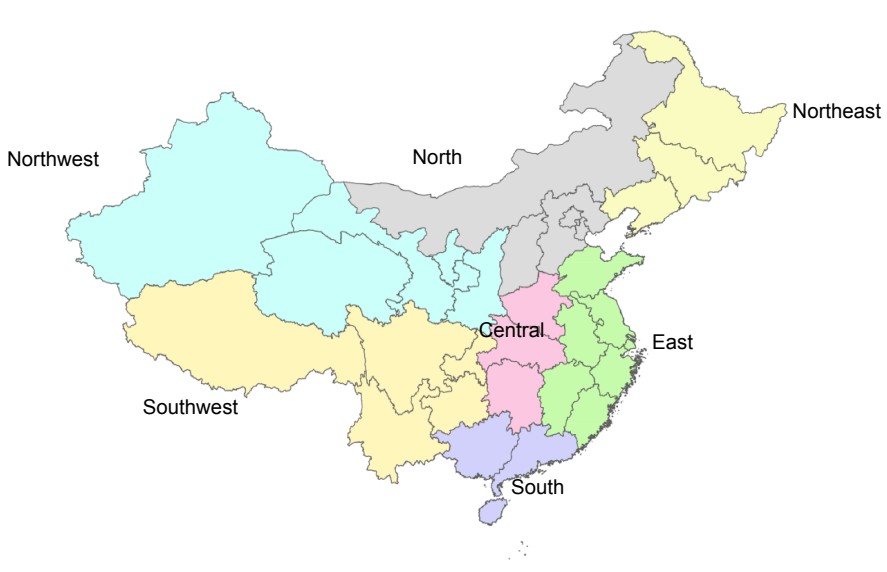

**Figure 1.** Seven regions in China used for analysis in this paper





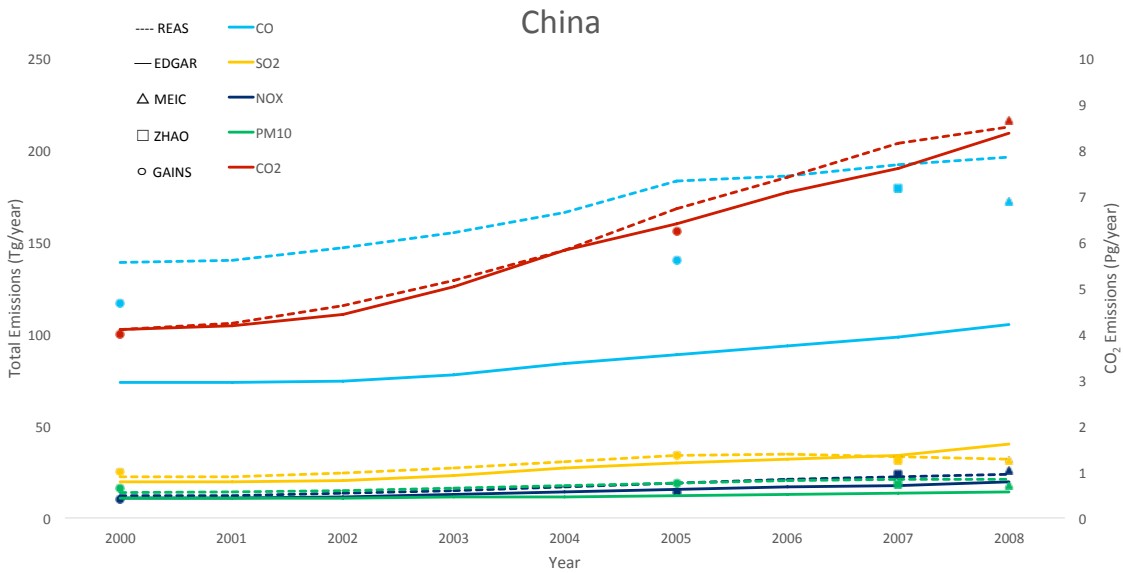

**Figure 2.** National total emissions estimates for CO, $SO_2$, $NO_x$, $PM_{10}$, and $CO_2$ estimated by REAS, EDGAR, MEIC, Zhao Yu, and GAINS between 2000 and 2008.




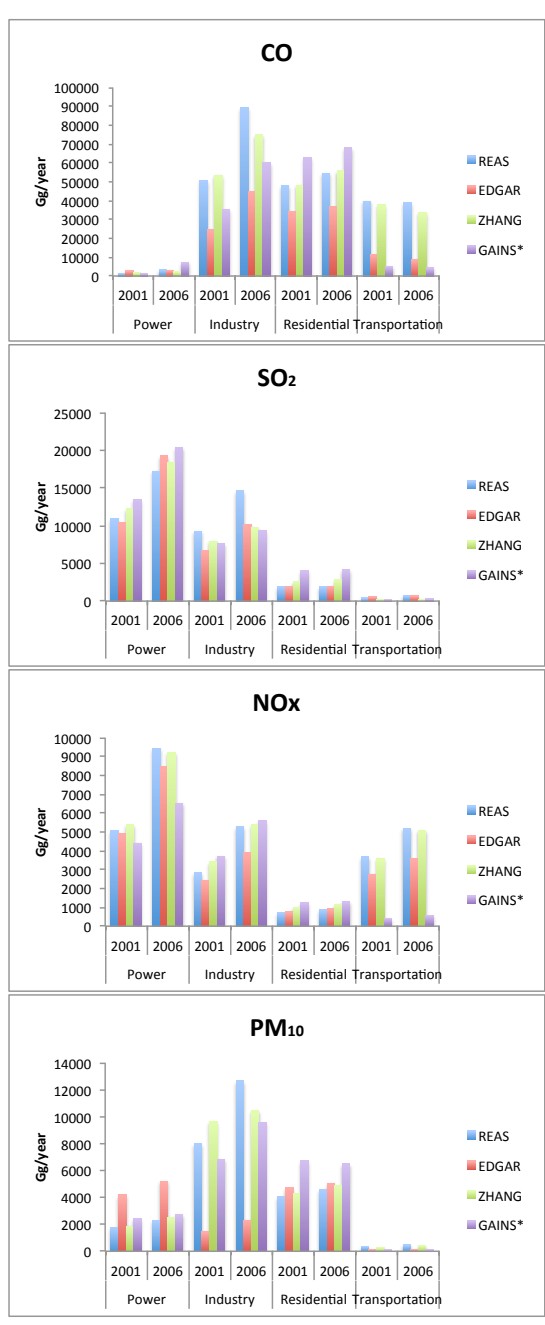

**Figure 3.** National emissions by source sector for four species. Zhang refers to Zhang et al. (2009). *GAINS
values are for 2000 and 2005.





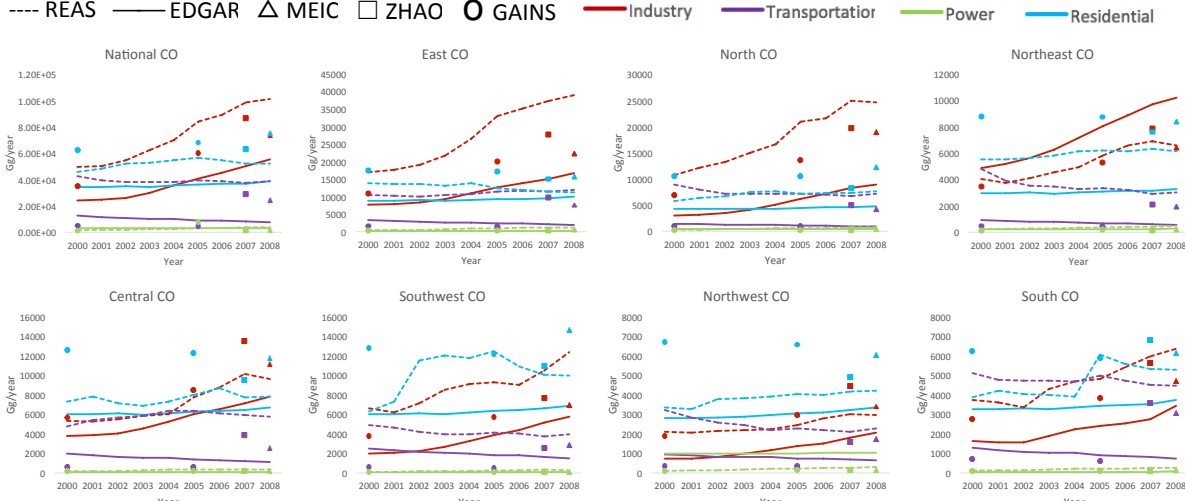

**Figure 4.** National and regional total emissions for CO for four different source sectors (industry, transportation, power, and residential) estimated by REAS, EDGAR, MEIC, Zhao Yu, and GAINS between 2000 and 2008.

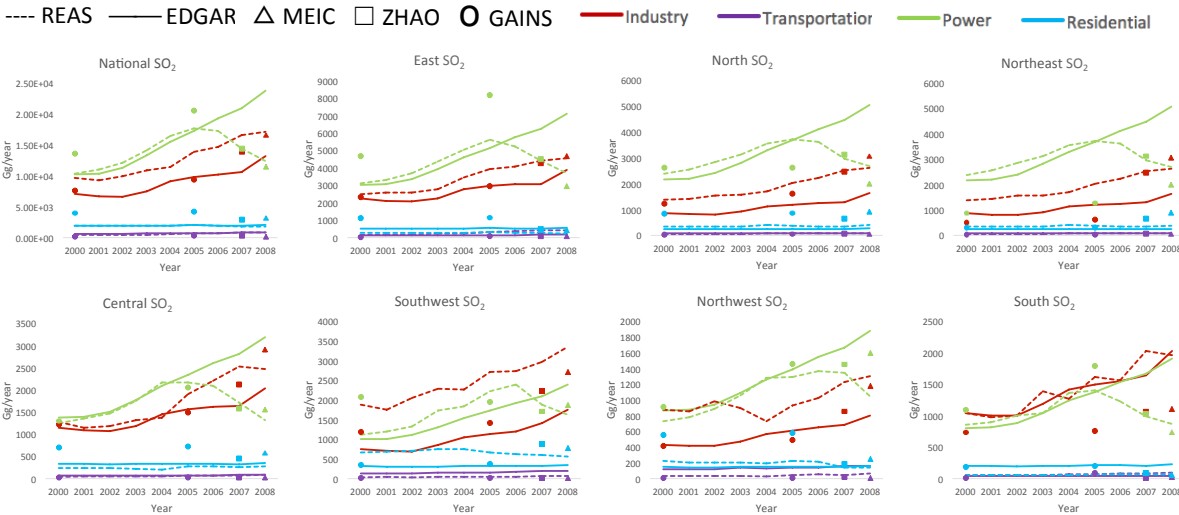

**Figure 5.** National and regional total emissions for SO$_2$ for four different source sectors (industry, transportation, power, and residential) estimated by EREAS, EDGAR, MEIC, Zhao Yu, and GAINS between 2000 and 2008.





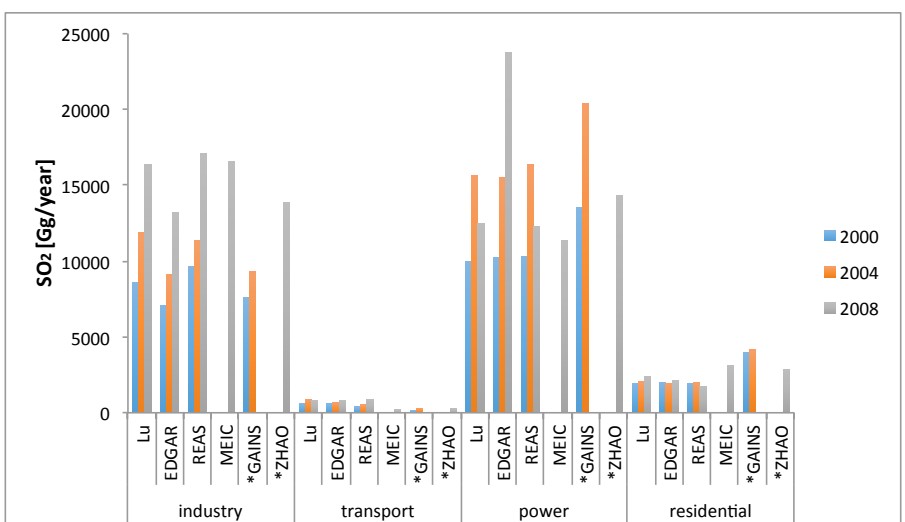

**Figure 6.** National SO$_2$ emissions estimates for four different source sectors (industry, transportation, power, and residential) estimated by EDGAR, REAS, MEIC, and Lu in 2000, 2004 and 2008. *GAINS values are for 2000 and 2005 and ZHAO's values are for 2007.

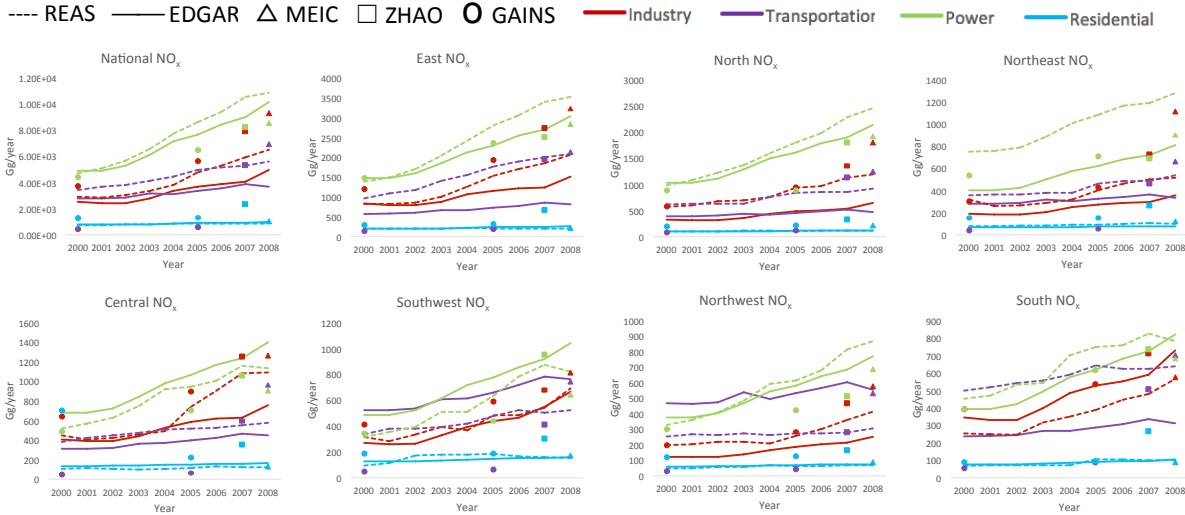

**Figure 7.** National and regional total emissions for NO$_x$ for four different source sectors (industry, transportation, power, and residential) estimated by REAS, EDGAR, MEIC, Zhao Yu, and GAINS between 2000 and 2008.





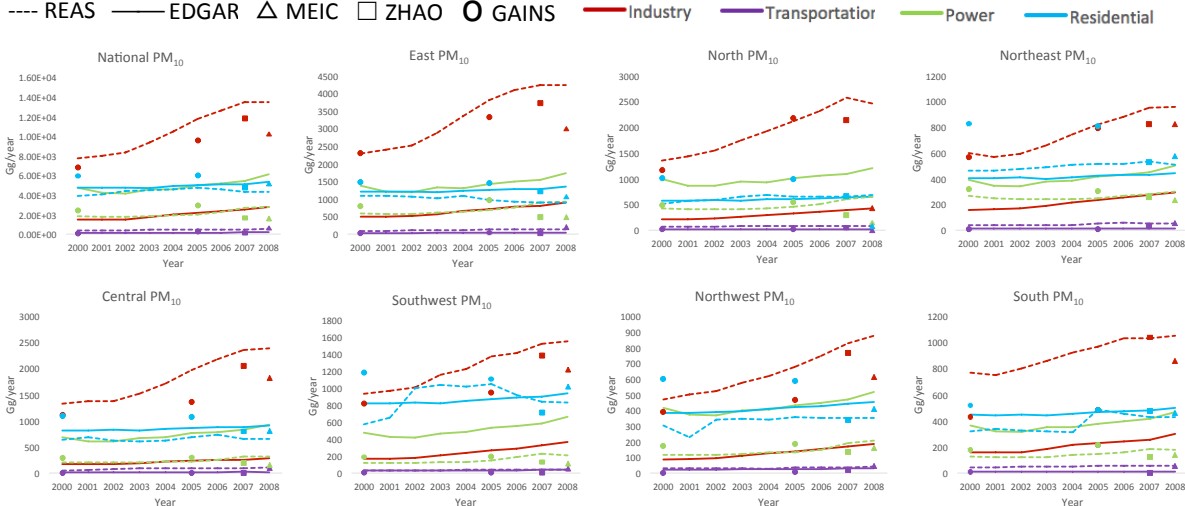

**Figure 8.** National and regional total emissions for $PM_{10}$ for four different source sectors (industry, transportation, power, and residential) estimated by EREAS, EDGAR, MEIC, Zhao Yu, and GAINS between 2000 and 2008.

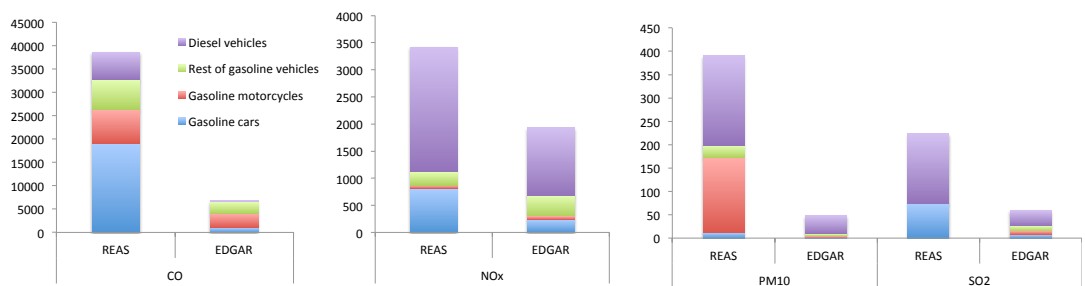

**Figure 9.** Transportation emissions estimated by REAS and EDGAR by fuel and vehicle types in 2008 in Gg/year. $SO_2$ emissions from the road-transportation sector are not calculated per different vehicle categories in REAS. Therefore, gasoline cars indicate total gasoline vehicles for $SO_2$ for REAS.




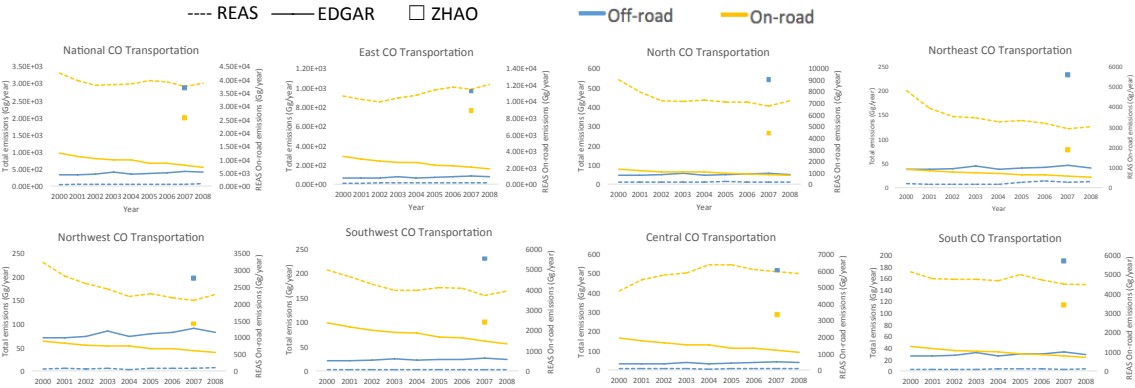

**Figure 10.** National and regional on-road and off-road transport sector emissions of CO estimated by EDGAR, REAS, and Zhao Yu, between 2000 and 2008.

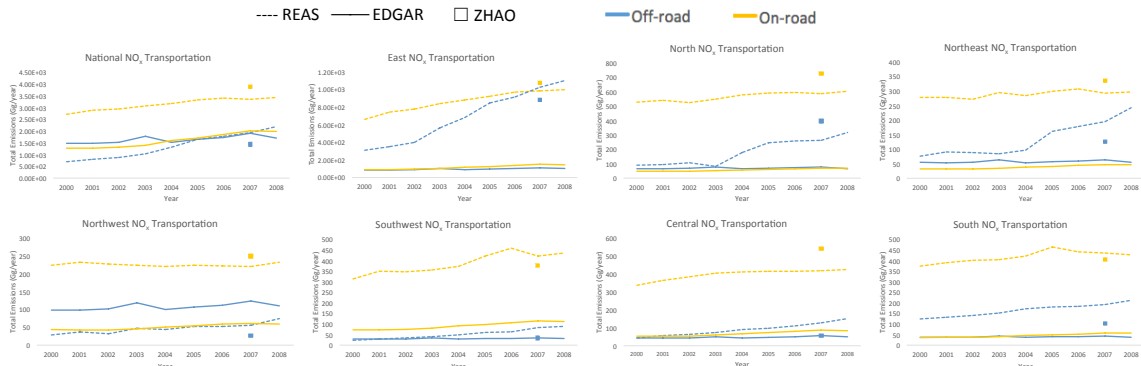

**Figure 11.** National and regional on-road and off-road transport sector emissions of $NO_x$ estimated by EDGAR, REAS, and Zhao Yu, between 2000 and 2008.





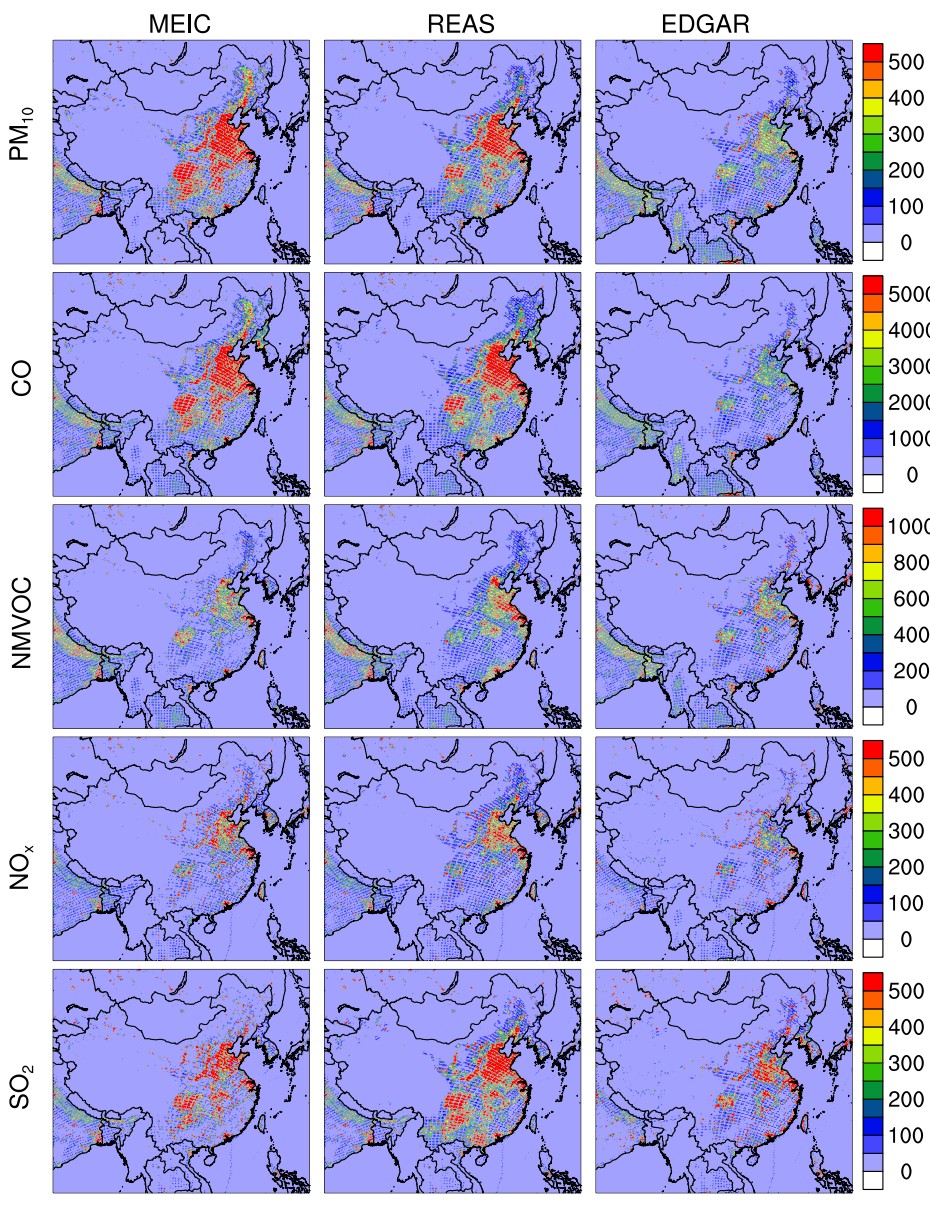

**Figure 12a.** Emissions of five pollutants (PM$_{10}$, CO, NMVOC, NO$_x$, and SO$_2$) in kg km$^{-2}$ month$^{-1}$ in 2008 January of the three emissions inventories.





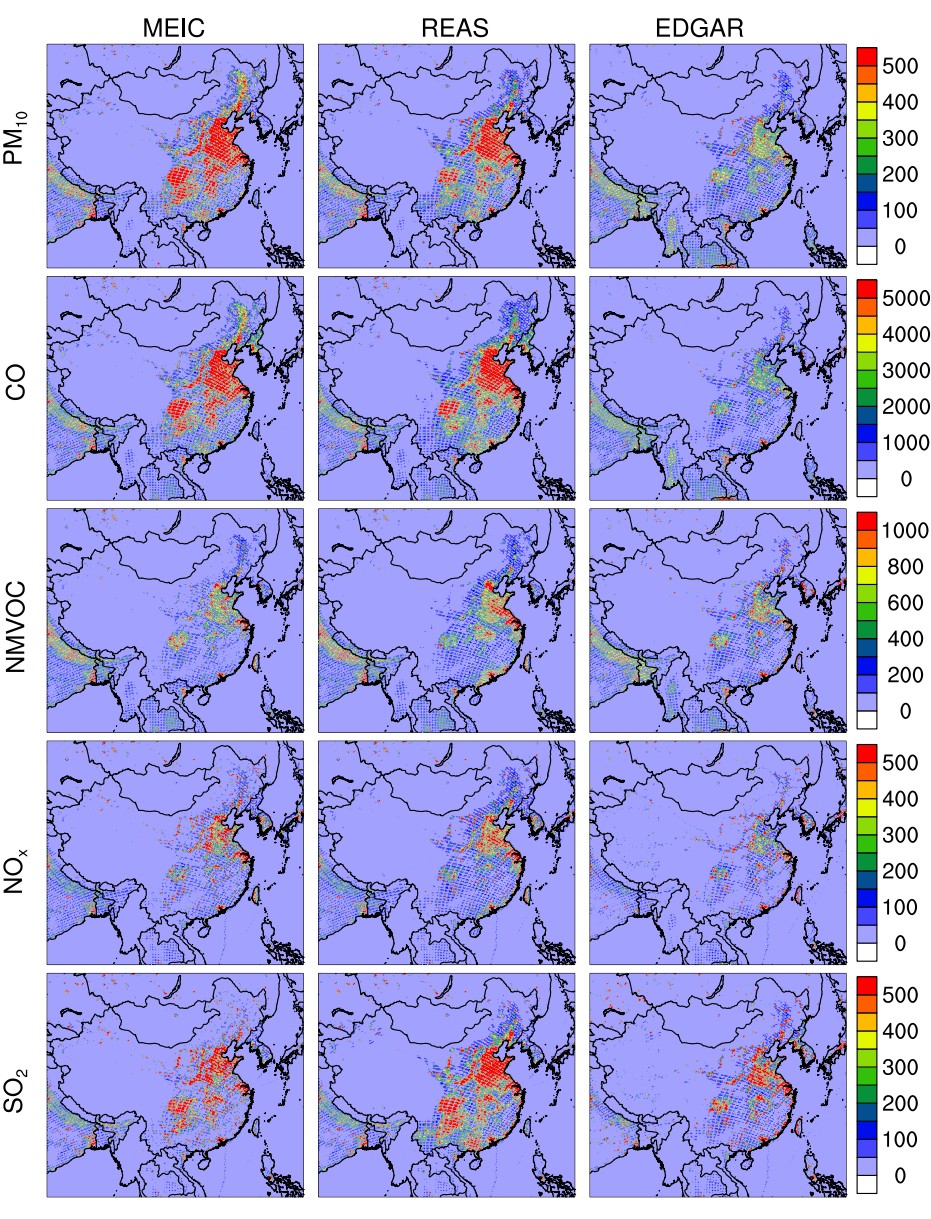

**Figure 12b.** Emissions of five pollutants (PM$_{10}$, CO, NMVOC, NO$_x$, and SO$_2$) in kg km$^{-2}$ month$^{-1}$ in 2008 July of the three emissions inventories.





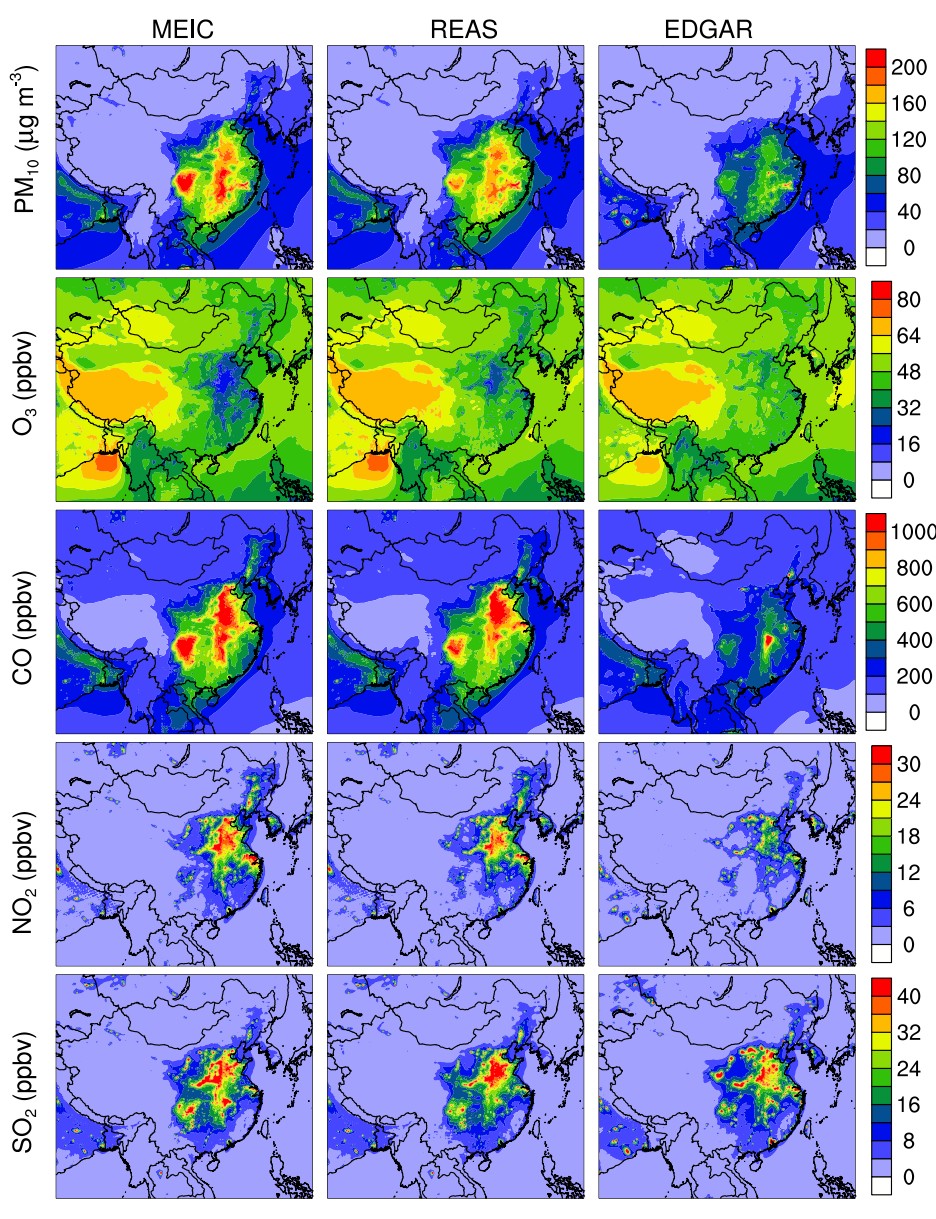

**Figure 13a.** Mixing ratios and concentrations of five pollutants in January using three emissions inventories.



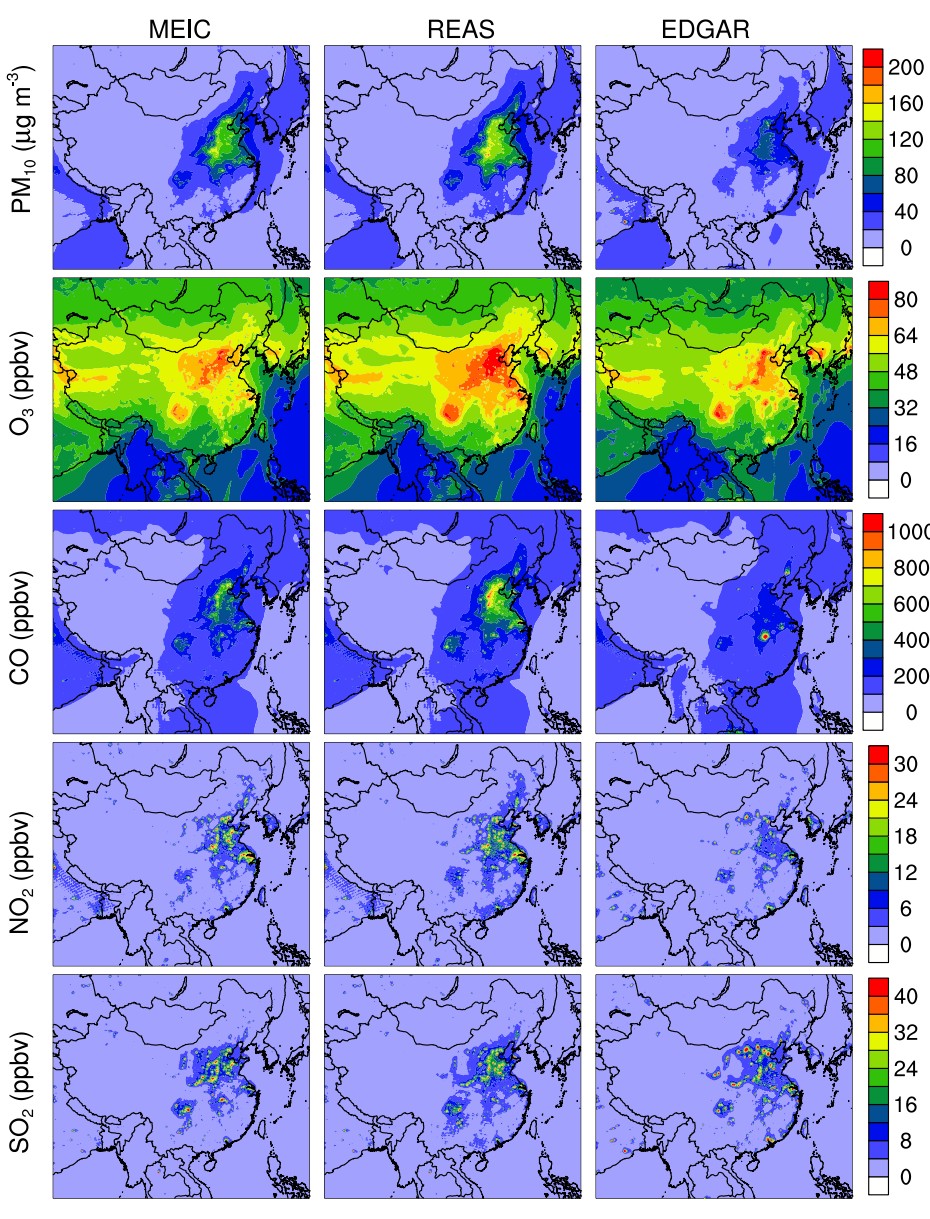

**Figure 13b.** Mixing ratios and concentrations of five pollutants in July using three emissions inventories.





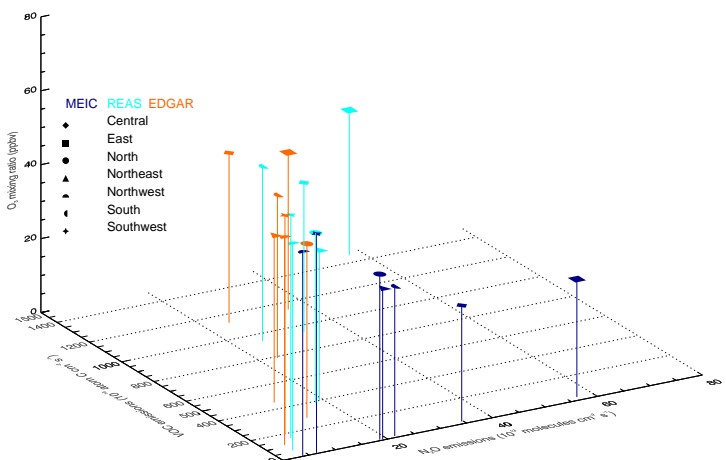

**Figure 14a.** Emissions of $NO_x$ and VOCs as well as $O_3$ mixing ratio in each region in January using three emissions inventories.

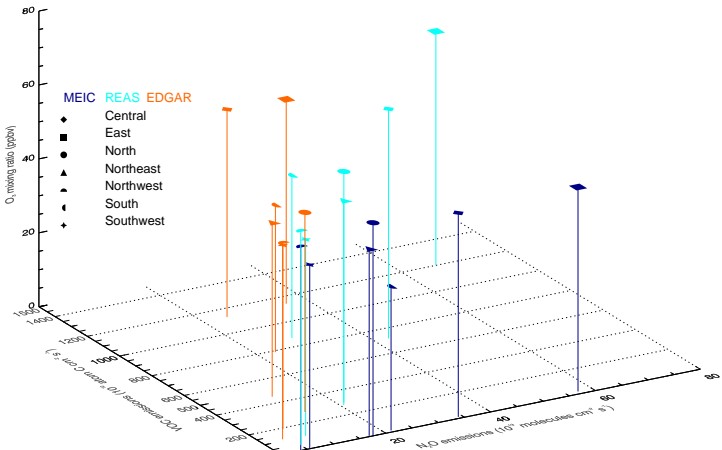

**Figure 14b.** Emissions of $NO_x$ and VOCs as well as $\tilde{O}_3$ mixing ratio in each region in July using three emissions inventories.





**Table 1.** Description of emissions inventories used for this study

| | Years | Source Sectors | Species | Horizontal Resolution | Coverage | Reference |
|---|---|---|---|---|---|---|
| REAS | 2000-2008 | power plants, combustible and non-combustible sources in industry, on-road and off-road sources in transportation, residential, agricultural, and other anthropogenic sources | $CO_2$, $SO_2$, CO, $PM_{10}$, $PM_{2.5}$, BC, OC, $NO_x$, $NH_3$, NMVOC, $CH_4$, $N_2O$ | 0.25° x 0.25° | East, Southeast, South & Central Asia. Asian part of Russia | Kurokawa et al., 2013 |
| EDGAR | 1970-2008 | energy, industrial processes, product use, agriculture, large scale biomass burning, and other anthropogenic sources | $CO_2$, $SO_2$, CO, $PM_{10}$, $NO_x$, $NH_3$, NMVOC, $CH_4$, $N_2O$ HFCs, $SF_6$, $NF_3$ | 0.1° x 0.1° | Global | EC-JRC/PBL, 2011 |
| MEIC | 2008, 2010 | power, industry, transportation, residential and agricultural sources | $CO_2$, $SO_2$, CO, $PM_{10}$, $NO_x$, NMVOC | 0.1° x 0.1° | China | www.meicmodel.org |
| ZHAO | 2000-2014 | power, combustible and non-combustible sources in industry, on-road and off-road sources in transportation and residential | $CO_2$, $SO_2$, CO, TSP, $PM_{10}$, $PM_{2.5}$, BC, OC, $NO_x$, Hg | N/A | China | Zhao et al., 2013b Zhao et al., 2015 Cui et al., 2015 Xia et al., 2016 |
| GAINS | 1990-2030 (5-yr increment, projection starting in 2015) | energy, domestic, industrial combustion and processes, road and non-road transportation and agriculture | $CO_2$, $SO_2$, CO, TSP, $PM_{10}$, $PM_{2.5}$, $PM_1$, BC, OC, $NO_x$, $NH_3$, VOC, $CH_4$, $N_2O$, F-gases | 0.5° x 0.5° | Amann et al., 2011 Global | Klimont et al., in review Klimont et al., in preparation |





**Table 2.** Source categorizations

| | EDGAR | REAS | | ZHAO | MEIC | GAINS | |
|---|---|---|---|---|---|---|---|
| Industry | Manufacturing industries and construction<br>Production of minerals<br>Production of chemicals<br>Production of metals<br>Production of pulp/paper/food/drink<br>Production of halocarbons and $SF_6$ | Combustible | Iron and steel<br>Chemical and petrochemical<br>Non-ferrous metal<br>Non-metallic minerals<br>Energy<br>Others | Industry | Industry | Combustible | Iron and steel<br>Pulp and Paper<br>Chemical<br>Non-ferrous metals<br>Non-metallic minerals<br>Other |
| | Refrigeration and air conditioning<br>Foam blowing<br>Fire extinguishers<br>Aerosols<br>F-gas as solvent<br>Semicondutor/electronics manufacturing<br>Electrical equipment<br>Other F-gas use<br>Solvent and other product use | Non-combustible | Pig iron<br>Crude steel<br>Iron steel others<br>Aluminum & Alumina<br>Copper<br>Zinc<br>Lead<br>Cement<br>Bricks<br>Lime<br>Coke ovens<br>Oil refinery<br>Other transformation<br>Sulphuric acid<br>Others | | | Processes | Pig iron<br>Coke ovens<br>Agglomeration plants<br>Steel<br>Rolling mills<br>Cast iron<br>Non-ferrous metals<br>Cement & Lime<br>Sulfuric acid<br>Nitric acid<br>Aluminium<br>Aluminium<br>Glass production<br>Fertilizer production<br>Brick manufacturing<br>Pulp and paper<br>Refineries<br>Others |
| Transportation | Domestic aviation<br>Road transportation<br>Rail transportation<br>Domestic navigation<br><br>Other transportation | | Cars<br>Buses<br>Light trucks<br>Heavy trucks<br>Motorcycles<br>Other vehicles<br>Domestic navigation<br>Railway & etc. | Light duty vehicles<br>Rural vehicles<br>Small gasoline engines<br>Heavy duty vehicles<br>Motorcycles<br>Machines<br>Inland shipping<br>Railway | Transportation | | Cars<br>Buses<br>Light duty vehicles<br>Heavy duty vehicles<br>Motorcycles<br>Mopeds<br>Domestic navigation<br>Railway & etc. |
| Power | Fugitive emissions from solid fuels<br>Fugitive emissions from oil and gas<br>Public electricity and heat production<br>Other energy industries<br>Non-energy use of lubricants/waxes ($CO_2$)<br>Fossil fuel fires | | Power plants | Power | Power | | Power plants<br>Diesel generators<br>Briquette production<br>Extraction and distribution of solid fuels<br>Extraction and distribution<br>of liquid & gaseous fuels |
| Residential | Residential and other sectors<br>Waste incineration | | Residential and other sectors | Residential and other sectors | Residential and other sectors | | Cooking and heating<br>Kerosene lighting<br>Waste (trash) burning |





**Table 3.** Number of power plants in each region within China

| Region | Number of coal power plants |
| --- | --- |
| East | 250 |
| North | 206 |
| Central | 86 |
| South | 78 |
| Northeast | 76 |
| Southwest | 66 |
| Northwest | 43 |

Source: Carbon Monitoring for Action





**Table 4.** Regional monthly mean concentrations of MEIC, REAS, and EDGAR and largest differences found within a region in WRF-Chem simulation in 2008

(a) January

| Regions | PM$_{10}$ ($\mu$g/m$^3$) | | | | O$_3$ (ppbv) | | | | SO$_2$ (ppbv) | | | | NO$_2$ (ppbv) | | | | CO (ppbv) | | | |
|---|---|---|---|---|---|---|---|---|---|---|---|---|---|---|---|---|---|---|---|---|
| | MEIC | REAS | EDGAR | diff | MEIC | REAS | EDGAR | diff | MEIC | REAS | EDGAR | diff | MEIC | REAS | EDGAR | diff | MEIC | REAS | EDGAR | diff |
| Central | 163 | 155 | 96 | 67 | 31 | 41 | 46 | 15 | 26 | 23 | 21 | 5.3 | 15 | 12 | 6.9 | 8.1 | 852 | 632 | 382 | 470 |
| East | 123 | 129 | 82 | 48 | 32 | 39 | 43 | 11 | 15 | 16 | 18 | 3.1 | 15 | 13 | 8.0 | 7.2 | 623 | 598 | 329 | 294 |
| North | 27 | 27 | 19 | 8.5 | 45 | 46 | 47 | 2.0 | 8.1 | 7.1 | 9.3 | 2.2 | 5.6 | 5.1 | 3.9 | 1.7 | 255 | 214 | 147 | 108 |
| Northeast | 29 | 25 | 14 | 15 | 41 | 41 | 46 | 4.5 | 4.2 | 5.1 | 5.9 | 1.7 | 6.6 | 6.5 | 3.4 | 3.3 | 259 | 242 | 165 | 94 |
| Northwest | 19 | 19 | 14 | 4.9 | 55 | 56 | 56 | 1.0 | 3.9 | 3.4 | 4.0 | 0.59 | 1.5 | 1.4 | 1.2 | 0.3 | 166 | 154 | 119 | 47 |
| South | 127 | 128 | 82 | 46 | 40 | 47 | 44 | 6.8 | 6.3 | 7.7 | 8.6 | 2.3 | 4.5 | 4.0 | 3.4 | 1.1 | 534 | 548 | 321 | 228 |
| Southwest | 42 | 37 | 23 | 19 | 59 | 60 | 59 | 1.2 | 5.1 | 4.9 | 3.4 | 1.7 | 1.5 | 1.4 | 1.3 | 0.13 | 284 | 242 | 156 | 128 |

(b) July

| Regions | PM$_{10}$ ($\mu$g/m$^3$) | | | | O$_3$ (ppbv) | | | | SO$_2$ (ppbv) | | | | NO$_2$ (ppbv) | | | | CO (ppbv) | | | |
|---|---|---|---|---|---|---|---|---|---|---|---|---|---|---|---|---|---|---|---|---|
| | MEIC | REAS | EDGAR | diff | MEIC | REAS | EDGAR | diff | MEIC | REAS | EDGAR | diff | MEIC | REAS | EDGAR | diff | MEIC | REAS | EDGAR | diff |
| Central | 64 | 72 | 37 | 34 | 55 | 62 | 56 | 6.9 | 7.3 | 7.0 | 7.1 | 0.27 | 5.4 | 6.7 | 4.0 | 2.7 | 263 | 300 | 224 | 77 |
| East | 56 | 64 | 36 | 28 | 55 | 63 | 55 | 8.1 | 6.9 | 8.4 | 9.4 | 2.5 | 7.8 | 8.7 | 5.0 | 3.7 | 247 | 321 | 192 | 129 |
| North | 39 | 33 | 21 | 13 | 58 | 63 | 54 | 8.5 | 4.2 | 4.5 | 5.6 | 1.3 | 2.6 | 3.0 | 2.0 | 1.0 | 178 | 212 | 130 | 82 |
| Northeast | 39 | 33 | 21 | 12 | 51 | 55 | 47 | 8.2 | 1.5 | 2.2 | 3.3 | 1.8 | 2.7 | 3.1 | 1.9 | 1.2 | 172 | 199 | 153 | 46 |
| Northwest | 8.4 | 8.8 | 6.3 | 2.5 | 55 | 58 | 53 | 4.8 | 1.4 | 1.3 | 1.6 | 0.22 | 0.57 | 0.78 | 0.61 | 0.21 | 94 | 95 | 90 | 5.3 |
| South | 19 | 21 | 17 | 3.8 | 39 | 44 | 40 | 5.0 | 2.4 | 4.3 | 5.2 | 2.9 | 2.9 | 4.0 | 3.1 | 1.1 | 170 | 185 | 156 | 29 |
| Southwest | 11 | 12 | 7.9 | 4.1 | 50 | 53 | 50 | 3.6 | 1.8 | 2.3 | 1.5 | 0.75 | 0.93 | 1.3 | 1.1 | 0.34 | 116 | 125 | 104 | 21 |