# Peer review of "Comparison of Emissions Inventories of Anthropogenic Air Pollutants in China"

_Atmospheric Chemistry and Physics, 2016_

## Referee Comment (RC1) · Anonymous Referee #1 · 2 Dec 2016

Review of Manuscript ACP-2016-888: 'Comparison of Emission Inventories of Anthropogenic Air Pollutants in China' by Eri Saikawa et al.

This manuscript presents five different emission inventories that cover the Asian region, specifically, China. It then compares and contrasts the differences in the inventories by air pollutant for China as a whole and broken down into a number of regions. Finally, three of the inventories are used to initiate some model runs to understand the implications of the differences outlined in the earlier sections. Overall, this is an informative paper, but rather straightforward. It would be good if the authors could dig into the differences a bit deeper and aim to understand the reasons behind the differences more than just presenting them. To a certain extent, I'm sure that the reasons behind these differences may not be easily discovered (if at all) since much of what is behind emission inventory construction is often not well documented, however, this paper really stays at the surface. Digging deeper would provide information that would be much more useful to modelers and others who will need to make decisions later as to which inventory to use and why, and if they are going to make modifications or not. I would recommend that this paper is published after revisions.

General comments:

-In section 2 each of the emission inventories are presented in a subsection. Please harmonize the descriptions in each of these subsections to cover, which regions are included, why the years were chosen as they were, which gridding/proxies/etc were important for each inventory.

Specifically, in section 2.3 for MEIC, the authors state that information for each Chinese province is included. Is that the same as the 33 sub-regions for REAS? Also there 'fine spatial resolution' is mentioned, can this be more quantitative to be able to compare? Later a 0.25x0.25 degree grid is mentioned, but this isn't even as high res as EDGAR – how does this fit together? How is the gridding for MEIC done?

Furthermore, for the Zhao inventory, why is 2007 used for the disaggregated emissions estimates when data for 2000-2014 are included and EDGAR, REAS, and MEIC provide 2008 data? Or even 2005 which would correspond to GAINS? Why not the whole time series?

-In section 3, can the authors address what is behind these estimates? Are some of them based on the same information? Completely different? When emission factors are discussed, is this information that can be included? Activity data, but same EF?

More specifically, on L249-255, some of these differences are hinted at, but no more detail is given. How do these mentioned EFs differ for the various sources?

-L359-364: can this text/discussion be elaborated a bit? This is exactly the type of understanding that is missing/typically not communicated in emission inventories and would be a very interesting addition.

-L394-398: 65% vs 38% is a pretty big difference. What is behind this difference? How close are the total amounts of PM10 emissions? Are the differences owing largely to the differences from other countries or the difference attributed to China mainly?

-section 4.1 & L476-477: what is driving these high off-road emissions for CO and NOx in the northwest? yes, the scales are different, but on-road tends to be higher in most other regions.

-section 5: the authors state that they chose 3 of the EI for the model simulations. But 5 were evaluated in the paper. I don't expect model simulations using all the EI, but a justification as to why those 3 were chosen should be added.

-L534-538 & L553-559: Here the authors compare the modeled to the observed values, and they are not even remotely close. Summer is better than winter, but still. I understand that models often over- or under-predict observed values, but this is a factor of 2 or more different. I also understand that model validation is not the point of this paper and it was more to demonstrate the implications of differences in EI, for which one might argue that the absolute concentration comparison to observed is not so important. However, while the models are described earlier, there are no references to model validation for the region, etc. Could something to at least reference this be included? It would be good to also at least acknowledge or try to explain this underestimation beyond just stating that it exists. Is this likely missing sources in the inventory? Poorly captured processes?

Specific comments:

-There are a number of words that are used incorrectly throughout the manuscript and should be replaced. Please do a search and replace, checking to make sure that the phrasing is still correct as written:

-discrepancy (definition: an illogical or surprising lack of compatibility or similarity between two or more facts) is used when difference would be much more appropriate.

-share; e.g., L333: 'Nationally, it shares 53, 33,.... of total SO2 emissions in REAS, ...' The industry sector does not 'share' anything. It should be written that SO2 emissions from industry contribute X amount to the national total.

-trends; this is not a language issue, but rather a scientific one. Trends are typically referring to a long time series of data for which a robust trend analysis has been done (e.g., with p-values, and a percent change per year over a minimum time period of 10-15 years or longer calculated). That is not how it is used here. I would suggest to avoid any confusion, that instances of 'trend' be replaced with 'change' since from what I can tell, it is always a percent change calculated from one year (e.g., 2000) to another year (2008), and that the concentrations of the years in between are not considered in this calculation. If this is not the case and an actual trend is calculated, this should be added to the methods section.

-L144-147: could these points mentioned in the text be added to Figure 3 where the years match to make the comparison easier? Also L173-174/L176?

-L187: The Schwartz et al 1994 reference is fine, but there are papers that would be more appropriate for health impacts of ozone.

-L209-210: This sentence doesn't make sense. The industry sector shares 51% of the difference in the estimates of what? Similarly, L241, '...sharing 43.7% of the difference in 2000 and 34.4% in 2008.' What does this mean? sharing the difference? please clarify.

-L320/Table 3: Are these the number of officially registered power plants? Are all officially registered? Is the data source reliable/are these numbers easy to get or is it likely that they are underestimated?

-In a number of cases, such as L346, percent changes are listed, but in many cases I think an absolute value change would be helpful because for example, in this case, the overall amount for SO2 emissions from residential sector is not high and this can be

pretty misleading then. -also L376-377: differences in sector listed as %, but how does this relate to the total? -again L479-480, how does this relate to absolute amounts? -L476: very dependent on the absolute values; although 258% seems like a huge amount. Please relate to the total to make it a more informative statement.

-L426-434: in the figure for PM, the REAS inventory shows a number of jumps for some regions. Can these be explained?

-L443-445: The text does not match the figure. The 'rest of gasoline' is not the majority share of any of the species. Nor is SO2 'non-existent' in REAS.

-L451-454: It seems odd to say we see significant differences in the CO, PM10, and SO2 emissions and then analyze the differences for different species, CO and NOx.

-why is it that in 4.1 and 4.2 that only 3 of the EI are included now? Justification?

-section 5.2: the authors discuss differences in concentration by region throughout this section, it would be good if they could add explicitly what these numbers represent. Are the values monthly average concentrations from all grid cells over the region? Or is it the maximum difference between monthly values for any single grid cell? Please clarify.

-L523-528: absolute amounts would help because the percents and concentration differences listed for CO are so huge, that it is then hard to relate the percents for the other species to concentrations, which are surely not similar to CO. In general, it would be good to mention table 4 which provides many of these concentrations much earlier in the section instead of only in the last 2 sentences.

-L550-551: this statement started out as relevant for NOx-VOC balance because of how these regimes affect ozone concentrations, and ended up as a blanket statement about how EI input is important. While the latter is true, it doesn't add much to the paper. Please avoid this and be more specific in the paper to really address the issue at hand.

-Figure 12b is never referenced or referred to in the text.

Minor edits:

-there are a number of small typos/english errors. I have specifically mentioned some here, but not all of them. Please try to read through this for such errors.

-L21: correct to '...for finding effective mitigation measures for reducing...'

-L25: correct to '...worst air quality countries in the world are located...'

-L44-47: here CO, NOx, SO2, and PM are mentioned, but NMVOCs are also mentioned in the abstract and subsequent text. Please add.

-L83: correct to '...was developed collaboratively between...'

-L84: correct to 'The inventory comprises emissions data from...'

-L191: correct to '...at the national level compared in Fig. 2 to all other species.'

-L199: correct to '...regardless of which inventory. Industry emissions contribute X, X, ... of the national total....'

-L280: add at the end of the sentence: '...in 2008, were emitted from this region.'

-L313-315: I would suggest to edit as follows: 'Up to this peak, REAS and EDGAR follow similar trajectories, but the SO2 emissions in the Central and the Northwest start to decrease in 2004, in 2005 in the South, East, and North, and in 2006 in the Northeast and the Southwest in REAS.'

-L317: define FGD

-L392: suggest to consider using 'patterns' or similar instead of 'trends'

-L413: do the authors mean 'reductions in EFs?' or are there reduction factors that are applied to emissions? Would be good to clarify either way.

-L418: replace 'troublesome' with 'difficult'

-L518: replace 'magnitudes' with 'concentrations' (or mixing ratios)

-Figure 5: there is a typo in REAS in the caption

[Figure]

---

## Referee Comment (RC2) · Anonymous Referee #2 · 6 Dec 2016

This manuscript examines 5 existing inventories of anthropogenic gases and aerosol in China. It compares emissions (CO, NOx, SO2, and PM) over national, regional, and sector level over 2000 to 2008. It then uses WRF-Chem to evaluate how the differences in emissions inventory influences air quality modeling. Overall, this is an informative paper and adds to the larger research discussion about uncertainty in emission inventories. However, many (but not all) of the comparisons between inventories are called out with simple comparisons with little effort to decompose the reasons behind the differences. In many sections, a deeper dive into why there are differences in the inventories would be really useful, similar to the discussion in L316 – 32 or L360 - 64, rather than just pointing out where differences occur. This is not always possible, as transparency and methodological documentation in inventories is often lacking, which the authors allude to, but even a discussion of why you can't explain the differences

would be helpful. Additionally, a discussion of how uncertainty varies over sectors and emission species would be helpful to put uncertainty in China inventories in context. I would recommend this paper for publication with revisions.

General Comments:

- In section 3, many of the sources sectors are compared across inventories as percent of total emissions. For example, (line 333) SO2 industry emissions have shares of 53,33,53,44, and 27% nationally for the 5 different inventories. This comparison is often somewhat misleading because the differences in other sectors, as well as aggregate totals, influence those percentages. For many of these comparisons, absolute emission values would be more informative.

- Manusrcipt is organized nicely, but writing style is very wordy. More concise writing style would aid in comprehension.

- Figure axes: many of the figure axes would benefit from formatting with commas or the use of Tg rather than Gg.

- This paper would benefit from a discussion or literature review of uncertainty in emissions inventories. Certain emissions species and sectors are more uncertain across the board in all countries. A discussion of how the differences in China inventories fit into that narrative (or don't) would be useful context.

- A summary discussion of the influence of activity data versus emissions factors in different sectors/regions would be helpful.

Specific Comments:

- Table 1: it looks like there is a reference, in the "Coverage" column for GAINS inventory

- Figure 1: The scale of the figure makes it difficult to see the differences between SO2, NOx, and PM10.

- The world "Total" in section title 3.1 and 3.2 is very misleading. The entire section is spent breaking down the national/regional TOTALS by sector.

- Figure 9 – label units of y axis

- L460 – 4: Why is Zhao estimate of off road estimates so much higher? – this is an example of where deeper discussion would be really useful.

- L153: please give a better discussion of figure 3

- L150: EDGAR doesn't "underestimate" CO emissions. It produces a smaller estimate than the other inventories. It may, infact underestimate CO emissions, but the analysis in this paper is not enough to assert that statement.

- L269-77: I'm not convinced that the ranking order of sectors "clearly illustrates" that emissions should be better constrained. Here (and elsewhere in the paper too) absolute differences (or percentages of sector totals) in inventory estimates would be more convincing than percent of total inventory value or ranks.

---

## Author Response (AR1)

We would like to thank the Editor and the reviewer for their comments on our paper. We appreciate the time that they have taken to read our manuscript and their comments and suggestions. Our replies to each of the referee comments are given below in blue.

Reviewer 1
This manuscript presents five different emission inventories that cover the Asian region, specifically, China. It then compares and contrasts the differences in the inventories by air pollutant for China as a whole and broken down into a number of regions. Finally, three of the inventories are used to initiate some model runs to understand the implications of the differences outlined in the earlier sections. Overall, this is an informative paper, but rather straightforward. It would be good if the authors could dig into the differences a bit deeper and aim to understand the reasons behind the differences more than just presenting them. To a certain extent, I'm sure that the reasons behind these differences may not be easily discovered (if at all) since much of what is behind emission inventory construction is often not well documented, however, this paper really stays at the surface. Digging deeper would provide information that would be much more useful to modelers and others who will need to make decisions later as to which inventory to use and why, and if they are going to make modifications or not. I would recommend that this paper is published after revisions.

Thank you for your comments. We have revised the paper based on your suggestion and addressed your comments below, as well as digging a bit deeper into the differences to understand the reasons behind the differences, as suggested.

General comments:

In section 2 each of the emission inventories are presented in a subsection. Please harmonize the descriptions in each of these subsections to cover, which regions are included, why the years were chosen as they were, which gridding/proxies/etc were important for each inventory.

We have harmonized the descriptions, as suggested by the reviewer. Section 2 now covers regions, years, sectors, and gridding proxies for each inventory.

Specifically, in section 2.3 for MEIC, the authors state that information for each Chinese province is included. Is that the same as the 33 sub-regions for REAS? Also there 'fine spatial resolution' is mentioned, can this be more quantitative to be able to compare? Later a 0.25x0.25 degree grid is mentioned, but this isn't even as high res as EDGAR – how does this fit together? How is the gridding for MEIC done?

REAS sub-regions include all the 31 sub-regions that are in the MEIC inventory, as well as Hong Kong and Macau. We have clarified this section by outlining the sub-regions better and we also removed "fine" from the spatial resolution in the MEIC description, as we agree with the reviewer's comment. The section 2.3 now reads as follows:
MEIC is an inventory developed at Tsinghua University, Beijing, China, and provides source sector information for the 31 Chinese sub-regions (all those included in the REAS,

except the two special administrative regions: Hong Kong and Macau) for 2008 and 2010 (Li et al., 2014; Zheng et al., 2014; Li et al., 2015; Liu et al., 2015). The MEIC model has a flexible spatial and sectoral resolution and allows for gridding of the emission product into user-specific grid including 0.25° longitude x 0.25° latitude horizontal resolution, as well as coarser grids. The emissions source sectors provided are power plants, industry, transport, residential and agricultural sources. Important proxy data for gridding of emissions includes population, roads, and power plants.

Furthermore, for the Zhao inventory, why is 2007 used for the disaggregated emissions estimates when data for 2000-2014 are included and EDGAR, REAS, and MEIC provide 2008 data? Or even 2005 which would correspond to GAINS? Why not the whole time series?

Thank you for this suggestion. We have tried to include as much as possible but 2007 was the only year where the data were disaggregated by the source sector. We therefore present the national total values for all species for 2000-2008 and include the disaggregated emissions estimates for 2007. We clarified this by the following text: A national emissions inventory for the 2000-2008 period was developed at Nanjing University (Zhao et al., 2008) and includes disaggregated information at the source sector and provincial levels for the year 2007.

-In section 3, can the authors address what is behind these estimates? Are some of them based on the same information? Completely different? When emission factors are discussed, is this information that can be included? Activity data, but same EF? More specifically, on L249-255, some of these differences are hinted at, but no more detail is given. How do these mentioned EFs differ for the various sources?

We now compare the net emission factors among EDGAR, REAS, MEIC, and GAINS, as well as the fuel data for the four source sectors and vehicle numbers for a few specific vehicle categories for the road transport sector. It was not possible to obtain information from all inventories and for all sectors as we had liked but we did our best to give as much detail as we possibly could. We have changed the section 3 significantly and a part of 3.1 reads as follows:
"Fig. 2 illustrates China's national total emissions for the four air pollutant species of our interest (CO, $SO_2$, NOx, and $PM_{10}$) as well as $CO_2$ estimated by REAS, EDGAR, MEIC, ZHAO, and GAINS, between 2000 and 2008, along with other published study estimates. We also used one million Monte Carlo samples from all emissions inventories, sector by sector, to create a composite emissions estimates for each species. For the inventories that provided a standard deviation or uncertainty, we used the information and assumed either a normal or log-normal distribution based on the information provided. If such information was not available, we used the relative uncertainty percentage provided by REAS to estimate standard deviation and assumed normal distribution.

We find the largest difference, ranging 65-94 Tg/year (87-106%), between REAS and EDGAR emissions estimates for total CO in China with REAS exceeding EDGAR throughout the 2000-2008 time period (Fig. 2). We further find that the major sectors

leading to the differences are industry and transport (Fig. 3). Indeed, between REAS and EDGAR, 38% of the difference in national total CO emissions stems from the industry sector in 2000. By 2008, the industry sector contributes 51% of the difference in their estimates.

What brings such a large difference from the industry sector? Coal combustion plays a large role in CO emissions from this sector in the REAS estimate and 98.6% of the combustible industrial emissions are due to coal in 2008. The comparison of fuel use statistics among REAS, EDGAR, and GAINS for 2000 (Fig. 4) and net emission factors per sector among REAS, EDGAR, GAINS, and MEIC (Fig. 5) are useful in understanding the reason behind the differences. The largest difference in fuel use is found for oil in the industry sector and a more than 9000 PJ/year difference exists between REAS and GAINS inventories. Coal use for industry also shows a more than 6000 PJ/year difference between REAS and GAINS (Fig. 4). However, considering that REAS and EDGAR show the largest difference and not REAS and GAINS for the Industrial CO emissions, it is clear that the difference in emission factors for industrial CO between REAS (2.2 ton CO/TJ) and EDGAR (1.1 ton CO/TJ) is the major reason for this difference, rather than the fuel use. Because emission factors are related to each technology type, penetration of the technology, uncontrolled emission factor and the emission reduction efficiency of each technology type, these factors all contribute to discrepancies. Obtaining estimates for CO is particularly troublesome because of many technology types that exist for emissions reduction. For the transport sector, estimated emissions by EDGAR are still lower than those of REAS (Fig. 3) even with its higher fuel use and emission factor, most likely because the modeling of superemitters have been omitted in EDGAR."

-L359-364: can this text/discussion be elaborated a bit? This is exactly the type of understanding that is missing/typically not communicated in emission inventories and would be a very interesting addition.

Thank you to your suggestion, we have expanded on the fuel use statistics, as illustrated earlier. Now we have new figures (Fig. 4 and 5), and we find that the difference in NOx emissions estimates are due to the difference in emission factors, rather than in fuel use estimates. We include the following paragraph in Section 3.1:
"The power emissions for NOx dominate the national total for REAS, EDGAR, and Zhang et al. (2009) (Fig. 3). 10.9 Tg yr$^{-1}$ (46%) and 10.2 Tg yr$^{-1}$ (51%) of the national NOx emissions are estimated to come from the power sector in REAS and EDGAR, respectively, in 2008. 6.5 Tg yr$^{-1}$ (47%) are estimated to come from the power sector in 2005 for GAINS. Streets et al. (2013) estimated power to be the dominant source sector, contributing 4.4 Tg yr$^{-1}$ (39% of NOx emissions) in 2000, followed by 2.8 Tg yr$^{-1}$ each (equal 25% contribution) from industry and transport. The national emissions inventories, however, do not show dominating power emissions for NOx. For MEIC, industrial emissions are estimated to be slightly higher than those from the power sector. For ZHAO, the two sources are similar in magnitude. 33% (36%) and 35% (35%) of the total emissions equaling 8.6 Tg yr$^{-1}$ (9.4 Tg yr$^{-1}$) and 8.3 Tg yr$^{-1}$ (7.9 Tg yr$^{-1}$) are estimated to come from the power (industry) sector in these two national inventories MEIC in 2008

and ZHAO in 2007, respectively. One of the possible reasons for this is due to the difference in emission factors among emission inventories (Fig. 5). MEIC estimates much higher emission factors for NOx emissions from the industry sector than from power, unlike other inventories that estimate the opposite (REAS and GAINS) or fairly close to each other (EDGAR).

-L394-398: 65% vs 38% is a pretty big difference. What is behind this difference? How close are the total amounts of PM10 emissions? Are the differences owing largely to the differences from other countries or the difference attributed to China mainly?

The total amounts of 2008 $PM_{10}$ emissions in China in 2008 in REAS and EDGAR estimates are 21.6 and 15.2 Tg/year, respectively. The total amounts of 2008 $PM_{10}$ emissions in the 22 Asian countries (including China) in 2008 in REAS and EDGAR estimates are 38.3 and 39.3 Tg/year. It is thus clear that although the total regional $PM_{10}$ emissions are quite similar in the two inventories, the estimates for China are not. We now include this in the manuscript as follows:
China's $PM_{10}$ emissions have been increasing rapidly and they contribute approximately 21.6 (15.2) Tg $yr^{-1}$ of 38.3 (39.3) Tg $yr^{-1}$ total $PM_{10}$ emissions from 22 Asian countries, including Afghanistan, Bangladesh, Bhutan, Nepal, Sri Lanka, India, Maldives, Pakistan, South Korea, North Korea, China, Japan, Singapore, Taiwan, Laos, Cambodia, Brunei, Myanmar, Philippines, Thailand, Vietnam, and Indonesia, in the REAS (EDGAR) estimate. There is a large difference between the estimates for China in the two inventories, although the regional total values are similar. Here, we only discuss primary emissions of $PM_{10}$, emitted directly from anthropogenic sources.

-section 4.1 & L476-477: what is driving these high off-road emissions for CO and NOx in the northwest? yes, the scales are different, but on-road tends to be higher in most other regions.

For CO, on-road emissions estimates are always higher than off-road in the Northwest in the three inventories. However, as we noted in L476-477, off-road NOx emissions are estimated to be higher than on-road in EDGAR, on average, by 57Gg/year in the Northwest. Neither REAS nor ZHAO estimate off-road to be higher in this region. This is because the railway activity assumed in EDGAR by coal and diesel locomotives in the region are much higher than the estimates by REAS and ZHAO. I have expanded this in the text as follows:
For the Northwest, EDGAR estimates larger emissions from off-road compared to on-road for NOx, which we do not see in either REAS or ZHAO. REAS estimates a higher growth rate for off-road emissions and their emissions estimates increase from 28.4 Gg $yr^{-1}$ in 2000 to 75.1 Gg $yr^{-1}$ in 2008, while EDGAR off-road emissions estimates show only a slight increase from 98.5 Gg $yr^{-1}$ to 110 Gg $yr^{-1}$ over the same time period. The large emissions differences are most likely due to much greater railway emissions by coal and diesel locomotives assumed in EDGAR inventory, compared to REAS, in this region.

-section 5: the authors state that they chose 3 of the EI for the model simulations. But 5 were evaluated in the paper. I don't expect model simulations using all the EI, but a justification as to why those 3 were chosen should be added.

We included three that had gridded emissions and we chose one global, one regional, and one national inventories to conduct simulations. These three also provided the maximum national total for most species (REAS), minimum national total for most species (EDGAR), and in between for most species (MEIC), as to provide a range in emissions estimates. We now have the following in the manuscript to justify our reasoning for these three inventories.
"We chose the three emissions inventories that provided gridded emissions and are targeted at different scales: EDGAR at global, REAS at regional, and MEIC at national. In addition, EDGAR estimates the lowest emissions for most species, whereas REAS estimates the highest and thus providing the range of air quality simulation from varying emissions. We then performed model simulations for January and July for 2008, using each of these inventories."

-L534-538 & L553-559: Here the authors compare the modeled to the observed values, and they are not even remotely close. Summer is better than winter, but still. I understand that models often over- or under-predict observed values, but this is a factor of 2 or more different. I also understand that model validation is not the point of this paper and it was more to demonstrate the implications of differences in EI, for which one might argue that the absolute concentration comparison to observed is not so important. However, while the models are described earlier, there are no references to model validation for the region, etc. Could something to at least reference this be included? It would be good to also at least acknowledge or try to explain this underestimation beyond just stating that it exists. Is this likely missing sources in the inventory? Poorly captured processes?

Thank you for this and we realize that we did not explain that this underestimation is mainly due to turning dust off in the model. We conducted a simulation without including dust in order to focus on the differences on air quality due to the different gridded emissions inputs. However, this method has led to a much larger difference in modeled values from the observed values. We have previously validated the model using dust in a paper by Zhong et al. (2015) and we make this clearer in this revised manuscript. The revision reads as follows:
"In order to focus on differences in air quality due to differing anthropogenic emissions estimates of gaseous pollutants and PM, we did not include dust in the model simulations in this study.

The model simulation including dust has been validated with existing measurements for the year 2007 in Zhong et al. (2015) and here we focus on differences in air quality simulation due to differing gridded anthropogenic emissions inputs."

Specific comments:

-There are a number of words that are used incorrectly throughout the manuscript and should be replaced. Please do a search and replace, checking to make sure that the phrasing is still correct as written:

-discrepancy (definition: an illogical or surprising lack of compatibility or similarity between two or more facts) is used when difference would be much more appropriate.

-share; e.g., L333: 'Nationally, it shares 53, 33,.... of total SO2 emissions in REAS, ...' The industry sector does not 'share' anything. It should be written that SO2 emissions from industry contribute X amount to the national total.

-trends; this is not a language issue, but rather a scientific one. Trends are typically referring to a long time series of data for which a robust trend analysis has been done (e.g., with p-values, and a percent change per year over a minimum time period of 10-15 years or longer calculated). That is not how it is used here. I would suggest to avoid any confusion, that instances of 'trend' be replaced with 'change' since from what I can tell, it is always a percent change calculated from one year (e.g., 2000) to another year (2008), and that the concentrations of the years in between are not considered in this calculation. If this is not the case and an actual trend is calculated, this should be added to the methods section.

Thank you for these corrections. We have changed our manuscript to make sure that the words we use are correct.

-L144-147: could these points mentioned in the text be added to Figure 3 where the years match to make the comparison easier? Also L173-174/L176?

Since these are the national total estimates and not the sector total, it is not possible for us to include these values in Figure 3. Instead, we created a new national total figure (Fig. 2) and included these values as well.

-L187: The Schwartz et al 1994 reference is fine, but there are papers that would be more appropriate for health impacts of ozone.

Thank you for this suggestion. We agree and we have inserted other papers, including Mudway and Kelly (2000) and Levy et al. (2005), which are more appropriate for health impacts of ozone. The revised manuscript now reads as follows:
"Atmospheric CO is mainly a result of incomplete combustion of fossil fuels and biofuels and exposure to ambient CO is harmful to human health (Aronow and Isbell1973; Stern et al., 1988; Allred et al., 1989; Morris et al., 1995). CO emissions are also important precursors to the formation of tropospheric $O_3$, which also has harmful human health impacts, including increased asthma exacerbations, decreased pulmonary function, and increased mortality (Schwartz et al., 1994; Mudway and Kelly 2000; Levy et al., 2005)."

-L209-210: This sentence doesn't make sense. The industry sector shares 51% of the difference in the estimates of what? Similarly, L241, '...sharing 43.7% of the difference in 2000 and 34.4% in 2008.' What does this mean? sharing the difference? please clarify.

Sorry for the confusion. We have revised the sentence to read as follows in the manuscript:
"Indeed, between REAS and EDGAR, 38% of the difference in national total CO emissions stems from the industry sector in 2000. By 2008, emissions difference in the industry sector contributes 51% of the total emissions difference for CO emissions in China."
"The third largest CO source and the source sector with the second largest difference after industry is transport, contributing 43.7% (34.4%) of the total difference in 2000 (2008)."

-L320/Table 3: Are these the number of officially registered power plants? Are all officially registered? Is the data source reliable/are these numbers easy to get or is it likely that they are underestimated?

This is based on the power plants listed in the Carbon Monitoring for Action (CARMA) database (http://carma.org/). This is the most transparent and most recent data available in terms of power plants and is used as a proxy for all inventories we compared in this paper. It is possible that they are underestimated but we do not have a better source to compare this number.

-In a number of cases, such as L346, percent changes are listed, but in many cases I think an absolute value change would be helpful because for example, in this case, the overall amount for SO2 emissions from residential sector is not high and this can be pretty misleading then.

We have changed the percentages to absolute values throughout the manuscript, based on the reviewer's suggestion. The sentence now reads as follows:
The residential sector emissions difference in the Southwest between EDGAR and REAS estimates have decreased from 354 Gg/year in 2000 to 215 Gg/year in 2008.

-also L376-377: differences in sector listed as %, but how does this relate to the total?

As mentioned above, we have changed the percentages to the absolute values, based on the reviewer's suggestion. The sentence now reads as:
In the South, Northwest, and Southwest, the difference in the transport sector emissions (percentage) among the inventories can also be as high as 560 (67%), 491 (72%), and 601 (83%) Gg/year, respectively.

-again L479-480, how does this relate to absolute amounts?

Same as above. The revised text reads as follows:

REAS estimates a higher growth rate for off-road emissions and their emissions estimates increase from 28.4 Gg yr$^{-1}$ in 2000 to 75.1 Gg yr$^{-1}$ in 2008, while EDGAR off-road emissions estimates only increase from 98.5 Gg yr$^{-1}$ to 110 Gg yr$^{-1}$ over the same time period.

-L476: very dependent on the absolute values; although 258% seems like a huge amount. Please relate to the total to make it a more informative statement.
Same as above. The current text now reads as follows:
For the East, REAS estimates an increase from 307 Gg/year to 1100 Gg/year in off-road emissions between 2000 and 2008.

-L426-434: in the figure for PM, the REAS inventory shows a number of jumps for some regions. Can these be explained?

The jumps we believe the reviewer indicated are the following:
1. the increase in Southwest from 2001 to 2002
2. the change in Northwest in 2000, 2001 and 2002
3. the increase in South from 2004 to 2005

The first jump is mainly due to the fuelwood consumption in Sichuan province within the Southwest region. The second jump for 2000/2001 is due to the change in fuelwood consumption in Shaanxi province and the change in crop residue consumption in Xinjiang province in 2001/2002 within the Northwest region. The third jump is due to the change in fuelwood and crop residue consumption in Guanxi province between 2004 and 2005 in the South region.

-L443-445: The text does not match the figure. The 'rest of gasoline' is not the majority share of any of the species. Nor is SO2 'non-existent' in REAS.

We are very sorry for the error. We have realized the mistake in the figure and forgot to update the text. Now the revised text reads as follows:
The majority of emissions (85% and 83%) come from gasoline vehicles in REAS and GAINS and almost all (97%) in EDGAR for CO. On the other hand, a significant contribution (67%, 65%, and 75%) comes from diesel vehicles for NOx in REAS, EDGAR, and GAINS, respectively. For PM$_{10}$, while REAS and GAINS estimates 390 Gg/year and 542 Gg/year, respectively, EDGAR only estimates 48 Gg/year. On-road SO$_2$ emissions also show a large difference between EDGAR (60 Gg/year) and REAS and GAINS (148 Gg/year and 200 Gg/year).

-L451-454: It seems odd to say we see significant differences in the CO, PM10, and SO2 emissions and then analyze the differences for different species, CO and NOx.

Yes, we agree and we now analyze these all species in more detail now in Section 4.

-why is it that in 4.1 and 4.2 that only 3 of the EI are included now? Justification?

Not all inventories have the information for on-road and off-road available and we compared with the four (we now include GAINS data in addition to REAS, EDGAR, and ZHAO) that we were able to collect. We were unable to obtain information from the MEIC inventory.

-section 5.2: the authors discuss differences in concentration by region throughout this section, it would be good if they could add explicitly what these numbers represent. Are the values monthly average concentrations from all grid cells over the region? Or is it the maximum difference between monthly values for any single grid cell? Please clarify.

Yes, these are monthly average concentrations from all grid cells over the region. Now we have inserted the following to clarify this in the manuscript:
For CO, both simulations using REAS and MEIC result in higher mixing ratios than when using EDGAR. We quantified the regional monthly mean of each simulation by averaging all grid cells in each region, as illustrated in Table 4. The REAS and MEIC regional monthly means are 270-470 ppbv (169-194 ppbv) higher in the polluted area in the Central (the East) region, than the EDGAR simulation. For $NO_2$, the largest differences in regional monthly mean occur between simulations using EDGAR and MEIC emissions, mainly in the Central (8.1 ppbv), followed by the East (7.2 ppbv) and the Northeast (3.3 ppbv). These regions are where the differences in emissions are the largest as well. For $SO_2$, both simulations using REAS and MEIC show differences in monthly mean less than 30% in most regions compared to those with EDGAR emissions, except in the Southwest, where REAS and MEIC estimates are 1.5 and 1.7 ppbv higher, respectively, than EDGAR estimates.

-L523-528: absolute amounts would help because the percents and concentration differences listed for CO are so huge, that it is then hard to relate the percents for the other species to concentrations, which are surely not similar to CO. In general, it would be good to mention table 4 which provides many of these concentrations much earlier in the section instead of only in the last 2 sentences.

We have changed the percentage to the absolute differences. We have also made changes to the section, such that Table 4 is mentioned much earlier in the section. The revision to this paragraph was stated as an answer to the previous question.

-L550-551: this statement started out as relevant for NOx-VOC balance because of how these regimes affect ozone concentrations, and ended up as a blanket statement about how EI input is important. While the latter is true, it doesn't add much to the paper. Please avoid this and be more specific in the paper to really address the issue at hand.

Thank you for this suggestion. We have now changed the text to illustrate that constraining NOx and VOC emissions in the Central and East regions are essential for understanding mitigation measures for $O_3$ in the future. The revised text reads as follows: This result illustrates the importance of constraining NOx and VOC emissions in the East and Central regions in understanding the way to mitigate $O_3$ pollution for the future.

-Figure 12b is never referenced or referred to in the text.

We now reference the Figure in the text.

Minor edits:

-there are a number of small typos/english errors. I have specifically mentioned some here, but not all of them. Please try to read through this for such errors.

-L21: correct to '...for finding effective mitigation measures for reducing...'

Corrected. Thank you.

-L25: correct to '...worst air quality countries in the world are located...'

Corrected. Thank you.

-L44-47: here CO, NOx, SO2, and PM are mentioned, but NMVOCs are also mentioned in the abstract and subsequent text. Please add.

We do not conduct an in-depth NMVOCs analysis in this paper, and so we changed the text as following to clarify this point:
The purpose of this study is to analyze the differences among the existing emissions inventory estimates for China's anthropogenic gaseous and aerosol emissions and how they affect air quality simulations. We analyze the emissions of carbon dioxide ($CO_2$), carbon monoxide (CO), sulfur dioxide ($SO_2$), nitrogen oxides (NOx), non-methane volatile organic compounds (NMVOCs), and particulate matter with an aerodynamic diameter less than 10 um ($PM_{10}$). We first evaluate the differences among inventories at the national level between years 2000 and 2008 for $CO_2$, CO, $SO_2$, NOx, and $PM_{10}$ and produce composite emissions estimates, using Monte Carlo samplings. Second, we focus on four source sectors (industry, transport, power, and residential) in seven regions of China (the East, North, Northeast, Central, Southwest, Northwest and South) for CO, $SO_2$, NOx, and $PM_{10}$. Next, we analyze the emissions estimates in the transport sector in more detail. By disaggregating emissions into these source sectors and regions, we aim to understand where the differences occur and how we can better constrain emissions. We also use a chemical transport model, the Weather Research and Forecasting model coupled with Chemistry (WRF-Chem) to assess how the different emissions estimates affect air quality modeling results.

-L83: correct to '...was developed collaboratively between...'

Corrected.

-L84: correct to 'The inventory comprises emissions data from...'

Corrected.

-L191: correct to '...at the national level compared in Fig. 2 to all other species.'

Corrected.

-L199: correct to '...regardless of which inventory. Industry emissions contribute X, X, ... of the national total....'
Corrected.

-L280: add at the end of the sentence: '...in 2008, were emitted from this region.'

Added.

-L313-315: I would suggest to edit as follows: 'Up to this peak, REAS and EDGAR follow similar trajectories, but the SO2 emissions in the Central and the Northwest start to decrease in 2004, in 2005 in the South, East, and North, and in 2006 in the Northeast and the Southwest in REAS.'

Changed.

-L317: define FGD

The FGD is defined as "flue-gas desulphurization" in L. 36.

-L392: suggest to consider using 'patterns' or similar instead of 'trends'

Changed.

-L413: do the authors mean 'reductions in EFs?' or are there reduction factors that are applied to emissions? Would be good to clarify either way.

This refers to the reduction factors that are applied to emissions for certain technologies. Rather than stating "emissions reduction factor" we rephrased it as "removal efficiency of a certain technology" to make it clearer.

-L418: replace 'troublesome' with 'difficult'

Corrected.

-L518: replace 'magnitudes' with 'concentrations' (or mixing ratios)

Corrected.

-Figure 5: there is a typo in REAS in the caption

Corrected.

We would like to thank the Editor and the reviewer for their comments on our paper. We appreciate the time that they have taken to read our manuscript and their comments and suggestions. Our replies to each of the referee comments are given below in blue.

Reviewer 2
This manuscript examines 5 existing inventories of anthropogenic gases and aerosol in China. It compares emissions (CO, NOx, SO2, and PM) over national, regional, and sector level over 2000 to 2008. It then uses WRF-Chem to evaluate how the differences in emissions inventory influences air quality modeling. Overall, this is an informative paper and adds to the larger research discussion about uncertainty in emission inventories. However, many (but not all) of the comparisons between inventories are called out with simple comparisons with little effort to decompose the reasons behind the differences. In many sections, a deeper dive into why there are differences in the inventories would be really useful, similar to the discussion in L316 – 32 or L360 - 64, rather than just pointing out where differences occur. This is not always possible, as transparency and methodological documentation in inventories is often lacking, which the authors allude to, but even a discussion of why you can't explain the differences would be helpful. Additionally, a discussion of how uncertainty varies over sectors and emission species would be helpful to put uncertainty in China inventories in context. I would recommend this paper for publication with revisions.

General Comments:

- In section 3, many of the sources sectors are compared across inventories as percent of total emissions. For example, (line 333) SO2 industry emissions have shares of 53,33,53,44, and 27% nationally for the 5 different inventories. This comparison is often somewhat misleading because the differences in other sectors, as well as aggregate totals, influence those percentages. For many of these comparisons, absolute emission values would be more informative.

We have changed the percentages to absolute emission values, based on the reviewer's suggestion. This sentence now reads as follows:
Nationally, it contributes 13 (53%), 17 (33%), 17 (53%), 14 (44%), and 9.3 (27%) Tg yr$^{-1}$ of total $SO_2$ emissions in REAS, EDGAR, MEIC for 2008, ZHAO for 2007, and GAINS for 2005, respectively.

- Manusrcipt is organized nicely, but writing style is very wordy. More concise writing style would aid in comprehension.

We have revised the manuscript to make it more concise.

- Figure axes: many of the figure axes would benefit from formatting with commas or the use of Tg rather than Gg.

We have revised the figures to make the axes easier to read.

- This paper would benefit from a discussion or literature review of uncertainty in emissions inventories. Certain emissions species and sectors are more uncertain across the board in all countries. A discussion of how the differences in China inventories fit into that narrative (or don't) would be useful context.

We have inserted the discussion of uncertainty in emissions inventories as follows: The difference in global CO, $SO_2$, and $NO_x$ emissions estimates among inventories is 28%, 42%, and 17% in 2000, respectively (Granier et al 2011). China's uncertainty is much larger for CO and NOx and 90% of global $CO_2$ emissions uncertainty stems from China.

- A summary discussion of the influence of activity data versus emissions factors in different sectors/regions would be helpful.

We have included the discussion of the influence of fuel use statistics and emission factors nationally per sector and we also discuss emission factors and vehicle categories in more detail for the road transport sector. We have changed the section 3 significantly and a part of 3.1 reads as follows:
"Fig. 2 illustrates China's national total emissions for the four air pollutant species of our interest (CO, $SO_2$, NOx, and $PM_{10}$) as well as $CO_2$ estimated by REAS, EDGAR, MEIC, ZHAO, and GAINS, between 2000 and 2008, along with other published study estimates. We also used one million Monte Carlo samples from all emissions inventories, sector by sector, to create a composite emissions estimates for each species. For the inventories that provided a standard deviation or uncertainty, we used the information and assumed either a normal or log-normal distribution based on the information provided. If such information was not available, we used the relative uncertainty percentage provided by REAS to estimate standard deviation and assumed normal distribution.

We find the largest difference, ranging 65-94 Tg/year (87-106%), between REAS and EDGAR emissions estimates for total CO in China with REAS exceeding EDGAR throughout the 2000-2008 time period (Fig. 2). We further find that the major sectors leading to the differences are industry and transport (Fig. 3). Indeed, between REAS and EDGAR, 38% of the difference in national total CO emissions stems from the industry sector in 2000. By 2008, the industry sector contributes 51% of the difference in their estimates.

What brings such a large difference from the industry sector? Coal combustion plays a large role in CO emissions from this sector in the REAS estimate and 98.6% of the combustible industrial emissions are due to coal in 2008. The comparison of fuel use statistics among REAS, EDGAR, and GAINS for 2000 (Fig. 4) and net emission factors per sector among REAS, EDGAR, GAINS, and MEIC (Fig. 5) are useful in understanding the reason behind the differences. The largest difference in fuel use is found for oil in the industry sector and a more than 9000 PJ/year difference exists between REAS and GAINS inventories. Coal use for industry also shows a more than 6000 PJ/year difference between REAS and GAINS (Fig. 4). However, considering that REAS and EDGAR show the largest difference and not REAS and GAINS for the

Industrial CO emissions, it is clear that the difference in emission factors for industrial CO between REAS (2.2 ton CO/TJ) and EDGAR (1.1 ton CO/TJ) is the major reason for this difference, rather than the fuel use. Because emission factors are related to each technology type, penetration of the technology, uncontrolled emission factor and the emission reduction efficiency of each technology type, these factors all contribute to discrepancies. Obtaining estimates for CO is particularly troublesome because of many technology types that exist for emissions reduction. For the transport sector, estimated emissions by EDGAR are still lower than those of REAS (Fig. 3) even with its higher fuel use and emission factor, most likely because the modeling of superemitters have been omitted in EDGAR."

Specific Comments:

- Table 1: it looks like there is a reference, in the "Coverage" column for GAINS inventory

Corrected. Thank you.

- Figure 1: The scale of the figure makes it difficult to see the differences between SO2, NOx, and PM10.

We changed the figure so that the differences are much more visible and we have also included other inventory values to make the comparison easier.

- The world "Total" in section title 3.1 and 3.2 is very misleading. The entire section is spent breaking down the national/regional TOTALS by sector.

We changed the subtitles to be National Level Comparisons and Regional Level Comparisons.

- Figure 9 – label units of y axis

Corrected. Thank you.

- L460 – 4: Why is Zhao estimate of off road estimates so much higher? – this is an example of where deeper discussion would be really useful.

Thank you for this question. We were unfortunately unable to compare the data to answer this specific question and we hope to do so in the future research.

- L153: please give a better discussion of figure 3

Thank you for your suggestion. We have changed the Fig. 3 to a new one, incorporating more inventories and over the whole time period to make our points come across better. We provide a better discussion of this revised Fig. 3 in the revised Section 3.

- L150: EDGAR doesn't "underestimate" CO emissions. It produces a smaller estimate than the other inventories. It may, infact underestimate CO emissions, but the analysis in this paper is not enough to assert that statement.

This is a very good point and we have revised the paper and changed to "For the transport sector, estimated emissions by EDGAR are still lower than those of REAS (Fig. 3)"

- L269-77: I'm not convinced that the ranking order of sectors "clearly illustrates" that emissions should be better constrained. Here (and elsewhere in the paper too) absolute differences (or percentages of sector totals) in inventory estimates would be more convincing than percent of total inventory value or ranks.

We have changed the sentence as follows:
At the national level, CO emissions are ranked first by industrial, next by residential, then by transport, and power. At the regional level, however, this ranking of source sectors does not always hold and also changes over time. For Northwest, emissions from the residential sector are estimated to be the largest in all years in all inventories. In Southwest, REAS estimates higher industrial emissions (6.6 Tg yr$^{-1}$ in 2000 and 12.4 Tg yr$^{-1}$ in 2008) than residential emissions (6.3 Tg yr$^{-1}$ in 2000 and 9.9 Tg yr$^{-1}$ in 2008) but EDGAR estimates higher transportation emissions (2.5 Tg yr$^{-1}$) than industrial (2.0 Tg yr$^{-1}$) in 2000. Similarly, in the South, REAS estimates industry to be the largest source sector (6.4 Tg yr$^{-1}$) followed by residential (5.3 Tg yr$^{-1}$) and transportation (4.5 Tg yr$^{-1}$) in 2008, whereas EDGAR estimates residential to be the largest (3.7 Tg yr$^{-1}$), followed by industry as a close second (3.4 Tg yr$^{-1}$) and transport (0.73 Tg yr$^{-1}$) with much lower emissions than the other two in the same year. This clearly illustrates the importance of constraining emissions at the disaggregated levels.

Manuscript prepared for Atmos. Chem. Phys.
with version 2015/04/24 7.83 Copernicus papers of the LaTeX class copernicus.cls.
Date: 30 March 2017

**Comparison of Emissions Inventories of Anthropogenic Air Pollutants in China**

Eri Saikawa[1,2], Hankyul Kim[2], Min Zhong[1], Yu Zhao[3], Greet Janssens-Maenhout[4], Jun-ichi Kurokawa[5], Zbigniew Klimont[6], Fabian Wagner[7], Vaishali Naik[8], Larry W. Horowitz[8], and Qiang Zhang[9]

[1]Department of Environmental Sciences, Emory University, Atlanta, GA
[2]Rollins School of Public Health, Emory University, Atlanta, GA
[3]School of the Environment, Nanjing University, Nanjing, China
[4]European Commission, Joint Research Centre, Directorate of Energy, Transport and Climate, Via Fermi, 2749, 21027 Ispra (VA), Italy
[5]Asia Center for Air Pollution Research, 1182 Sowa, Nishi-ku, Niigata, Niigata, 950-2144, Japan
[6]International Institute for Applied Systems Analysis, Laxenburg, Austria
[7]Andlinger Center for Energy and the Environment, Princeton University, Princeton, NJ
[8]NOAA Geophysical Fluid Dynamics Laboratory, Princeton, NJ, USA
[9]Center for Earth System Science, Tsinghua University, Beijing, China

*Correspondence to:* Eri Saikawa (eri.saikawa@emory.edu)

**Abstract.** Anthropogenic air pollutant emissions have been increasing rapidly in China, leading to worsening air quality. Modelers use emissions inventories  to represent the temporal and spatial distribution of these emissions needed to estimate their impacts on regional and global air quality. However, large uncertainties exist in emissions estimates assessing

5  differences in these inventories is essential for better understanding of  air pollution over China. We compare five different emissions inventories estimating emissions of carbon dioxide ($CO_2$), carbon monoxide (CO), nitrogen oxides ($NO_x$), sulfur dioxide ($SO_2$), and particulate matter with an aerodynamic diameter of 10 $\mu$m or less ($PM_{10}$) from China. The emissions inventories analyzed in this paper include Regional Emissions inventory in ASia v2.1 (REAS); Multi-resolution

10  Emission Inventory for China (MEIC); Emission Database for Global Atmospheric Research v4.2 (EDGAR); the inventory by Yu Zhao (ZHAO); and the Greenhouse Gas and Air Pollution Interactions and Synergies (GAINS). We focus on the period between 2000 and 2008 during which  Chinese economic activities have more than doubled. In addition to  national totals, we also analyzed emissions from four source sectors (industry, transport, power, and residential) and

15  within seven regions in China (East, North, Northeast, Central, Southwest, Northwest, and South) and found that large disagreements  exist among the five inventories at disaggregated levels. These discrepancies lead to differences of 67$\mu$g/m$^3$, 15ppbv, and 470ppbv for monthly mean $PM_{10}$, $O_3$, and CO, respectively, in modelled regional concentrations in China. We also find that  all the inventory emissions estimates create a VOC-limited environment  and MEIC emis-

20  sions  lead to much lower $O_3$ mixing ratio in  East and Central China compared to the

simulations using REAS and EDGAR estimates, due to its low VOC emissions. Our results illustrate that a better understanding of Chinese emissions at more disaggregated levels is essential for finding  effective mitigation measures for reducing national and regional air pollution in China.

**1 Introduction**

25  Obtaining accurate emissions estimates for air pollutant species is important in Asia, where five of the worst air quality countries in the world  are located (Hsu et al., 2014). Emissions of ozone precursors, including nitrogen oxides ($NO_x \equiv NO + NO_2$) and carbon monoxide (CO), affect tropospheric ozone ($O_3$) mixing ratio at local, regional, and inter-continental scales (Fiore et al., 2009; West et al., 2009). In addition to the emissions of primary particulate matter (PM), those of

30  sulfur dioxide ($SO_2$) and $NO_x$ also affect PM concentrations on local and regional scales. Both surface $O_3$ and PM are linked to adverse health impacts (Dockery et al., 1993; Levy et al., 2001; Pope III et al., 2002), and $O_3$ also affects agricultural crop yields (Heck et al., 1983; Krupa and Manning, 1988; Avnery et al., 2011).

One key country in need of accurate emissions estimates is China, the largest emitter and the

35  biggest contributor to the uncertainty in the source and the magnitude of many of the air pollutant species. The difference in global CO, $SO_2$, and $NO_x$ emissions estimates among inventories is 28%, 42%, and 17% in 2000, respectively (Granier et al., 2011). China˘s uncertainty is much larger for CO and $NO_x$ and 90% of global $CO_2$ emissions uncertainty stems from China (Andres et al., 2014). Energy consumption has been steadily increasing in China but at the same time, the

40  implementation of emissions control measures, including the flue-gas desulphurization (FGD) in coal-fired power plants, has led to rapid changes in emission factors in recent decades (Xu, 2011; Zhang et al., 2012; Kurokawa et al., 2013). Several emissions inventories have been developed in the past, either specifically for China or for Asia (Streets and Waldhoff, 2000; Streets et al., 2003; Zhao et al., 2008; Klimont et al., 2009; Lu et al., 2010; European Commission, Joint Research Cen-

45  tre (JRC)/Netherlands Environmental Assessment Agency (PBL), 2011; Lei et al., 2011; Lu et al., 2011; Smith et al., 2011; Zhao et al., 2012; Kurokawa et al., 2013; Klimont et al., 2013) but none have assessed or compared emissions from different source sectors at more disaggregated scales than the national level.

The purpose of this study is to analyze the  differences among the existing emissions

50  inventor estimates for China's anthropogenic gaseous and aerosol emissions  and how they affect air quality simulations. We analyze the emissions of carbon dioxide ($CO_2$), carbon monoxide (CO), sulfur dioxide ($SO_2$), nitrogen oxides ($NO_x$), non-methane volatile organic compounds (NMVOCs), and particulate matter with an aerodynamic diameter less than 10 $\mu$m ($PM_{10}$). We first evaluate the differences among inventories at the national level between years 2000 and

55  2008 for $CO_2$, CO, $SO_2$, $NO_x$, and $PM_{10}$ and produce composite emissions estimates, using Monte

Carlo samplings. Second , we focus on four source sectors (industry, transport, power, and residential) in seven regions of China (the East, North, Northeast, Central, Southwest, Northwest and South) for CO, $SO_2$, $NO_x$, and $PM_{10}$. Next, we analyze emissions estimates in the transport sector in more detail. By disaggregating emissions into these source sectors and regions, we aim to under-

60    stand where the  differences occur and how we can better constrain emissions. We also use a chemical transport model, the Weather Research and Forecasting model coupled with Chemistry (WRF-Chem), to assess how the different emissions  estimates affect air quality modeling results.

The paper is organized as follows. Section 2 explains the emissions inventories that we have

65    compared. Section 3 analyzes the differences in emissions inventories first at the national level and then in seven regions within China. Section 4 compares transport sector emissions in depth. Section 5 describes the impact of the emissions inventories on air quality simulations. Section 6 presents a summary of results and suggested future research.

**2    Emissions Inventories**

70    In this study, we compare five existing emissions inventories at the national, regional, and source sector levels between years 2000 and 2008 (Table 1). The Regional Emission inventory in ASia version 2.1 (REAS) is a regional emissions inventory for most of the Asian countries including the East, Southeast, South, and Central Asia and the Asian part of Russia (Kurokawa et al., 2013). The Emission Database for Global Atmospheric Research version 4.2 (EDGAR) is a global emissions

75    inventory and includes major air pollutants from combustion and non-combustion sources (European Commission, Joint Research Centre (JRC)/Netherlands Environmental Assessment Agency (PBL), 2011). Multi-resolution Emission Inventory for China (MEIC, http://meicmodel.org/) is an inventory developed at Tsinghua University, Beijing, China, and provides national emissions estimates for 2008 and 2010. A national emissions inventory for the 2000-2008  period was developed at

80    Nanjing University (Zhao et al., 2008) and includes disaggregated information at the source sector and provincial levels for the year 2007. The Greenhouse Gas and Air Pollution Interactions and Synergies (GAINS, http://gains.iiasa.ac.at/models/index.html) model is a framework for analyzing co-benefits of reduction strategies  for air pollution and greenhouse gas sources globally  which provides estimates of emissions  including province-level

85    emissions from China (Amann et al., 2011). These five emissions inventories were developed using a similar methodology, where emissions were calculated as the product of activity data, such as fuel consumption or industrial production, emission factors of combustion or production technology, and penetration rate and emission reduction efficiency of emission controls (Zhao et al., 2014). Table 2 shows how emissions in each of the inventories are aggregated to the four primary source sectors

90    (industry, transport, power, and residential) that we analyze in this paper. They were grouped in this

way to be able to compare at the four source sector levels among the inventories, as this is how some of the inventories (i.e., MEIC) are structured. Here we explain each of the emissions inventories in more detail.

**2.1 REAS**

95  REAS was developed  collaboratively between the National Institute for Environmental Studies and Asia Center for Air Pollution Research, Japan (Kurokawa et al., 2013). The inventory comprises  emissions data from 30 Asian countries and regions, including China divided into 33 sub-regions (22 provinces, five autonomous regions, four municipalities, and two special administrative regions), between years 2000 and 2008 at a 0.25° longitude x 0.25° latitude horizon-
100  tal resolution. A previous version of REAS spanned a longer time period  and included projections of emissions (Ohara et al., 2007) but v2.1 is based on updated activity data and parameters. The emissions sources provided are power plants, combustible and non-combustible sources in industry, on-road and off-road sources in transport, and residential and others such as agricultural activities and evaporative sources. Important proxies for gridding include rural, urban, and
105  total populations, as well as road networks.

**2.2 EDGAR**

EDGAR was developed by the Joint Research Centre of the European Commission, in collaboration with the Netherlands Environmental Assessment Agency (European Commission, Joint Research Centre (JRC)/Netherlands Environmental Assessment Agency (PBL), 2011). This database incorpo-
110  rated experiences of the dataset EDGAR v3.2 FT2000 from Olivier et al. (2001). EDGAR is a gridded emissions inventory of greenhouse gases, air pollutants and aerosols that spans 1970 - 2008 at a 0.1° longitude x 0.1° latitude horizontal resolution. The source sectors provided are energy, industrial processes, product use, agriculture, waste, and other anthropogenic sources. Country emissions are compiled based on the International Energy Agency (IEA) energy statistics
115  . Emission factors are taken from the EMEP/EEA air pollutant emission inventory guidebook (European Environment Agency, 2013) and other scientific literature. Gridding of national total emissions is done using several types of proxy data (population, road, power plants, animals, crop) as described in Janssens-Maenhout et al. (2013).

120  ## 2.3 MEIC

MEIC is an inventory developed at Tsinghua University, Beijing, China, and provides source sector information for  the 31 Chinese sub-regions (all those included in the REAS, except the two special administrative regions: Hong Kong and Macau) for 2008 and 2010 (Li et al., 2014; Zheng et al., 2014; Li et al., 2015; Liu et al., 2015). The MEIC model has  a flexible

spatial and sectoral resolution and allows for gridding of the emission product into a user-specific grid including 0.25° longitude x 0.25° latitude horizontal resolution, as well as coarser grids. The emissions source sectors provided are power plants, industry, transport, residential and agricultural sources. Important proxy data for gridding of emissions includes population, roads, and power plants.

**2.4 ZHAO**

The inventory made at Nanjing University is a national inventory that estimates source sector emissions from all the 31 Chinese  sub-regions, the same as MEIC (Zhao et al., 2013b, 2015; Cui et al., 2015; Xia et al., 2016). The inventory includes the national-level data for 2000-2008  and we use the available disaggregated emissions estimates for 2007 for comparison. The sectors provided are industry (including cement, iron & steel, other industrial combustion, and other industrial processes), power, transport (including on-road and off-road), and residential. This inventory does not provide gridded emissions.

**2.5 GAINS**

The GAINS model was developed at the International Institute for Applied Systems Analysis and estimates global emissions, including those for the 31  sub-regions in China, as in MEIC and ZHAO, as well as Hong Kong and Macau, as in REAS (Amann et al., 2008; Klimont et al., 2009). The GAINS model calculates emissions estimates in five-year intervals from 1990 to 2050, with the projection starting in year 2015. It has a large number of source sectors including energy, domestic, industrial combustion and processes, road and non-road transport, and agriculture, for which activities originate from international and national statistics. It provides output in various formats and spatial resolution, including 0.5° latitude x 0.5° longitude horizontal grid. For this study, we use estimates from energy, domestic, transport, and industry sectors for the years 2000 and 2005, using the global dataset developed within the European Union project ECLIPSE (version V5a, http://www.iiasa.ac.at/web/home/research/researchPrograms/air/Global_emissions.html) (Klimont et al., 2016). Sectoral proxies used in Representative Concentration Pathways (RCP) and Global Energy Assessment (GEA), as well as population and selected industrial plant locations are used as important proxies for gridding.

**3 National and Regional Comparisons**

To better understand the  differences among anthropogenic emissions estimates of four air pollutant species, we first analyzed differences in national total emissions estimates between years 2000 and 2008. For each of the species, we further compared these estimates in seven different

regions (Fig. 1) for four source sectors separately. In the following sections, we first describe the  differences at the national level, and then at the regional level for each species.

**3.1 National  Level Comparisons**

160 Fig. 2 illustrates China's national total emissions for the four air pollutant species of our interest (CO, $SO_2$, $NO_x$, and $PM_{10}$) as well as $CO_2$ estimated by REAS, EDGAR, MEIC, ZHAO, and GAINS, between 2000 and 2008, along with other published study estimates. We also used one million Monte Carlo samples from all emissions inventories, sector by sector, to create a composite emissions estimate for each species. For the inventories that provided a standard deviation or uncertainty, we used

165 the information and assumed either a normal or log-normal distribution based on the information provided. If such information was not available, we used the relative uncertainty percentage provided by REAS for a sector for each species to estimate standard deviation and assumed normal distribution.

We find the largest  difference, ranging from 65 - 94 Tg/year (87 - 106%), between

170 REAS and EDGAR emissions estimates for total CO in China with REAS exceeding EDGAR throughout the 2000 - 2008 time period (Fig. 2).

175 ~~Streets et al. (2006) estimated 151 Tg/year for 2001. Zhao et al. (2012) estimated 173, 179, 179, and 167 Tg/year for the years 2005, 2006, 2007, and 2008, respectively, and Zhang et al. (2009) estimated 167 Tg/year for 2006. These are all well-aligned with the REAS estimates. Top-down estimates, optimizing the emissions using both observational data and the simulations from atmospheric chemical transport models, for the early 2000s were also high, ranging between 140 and 230~~

180  We further find that the major sectors leading to the differences are industry and transport (Fig. 3). Indeed, between REAS and EDGAR, 39% of the difference in national total CO emissions stems from the industry sector in 2000. By 2008, the emissions difference in the industry sector contributes 51% of the total emissions difference for CO emissions in China .

185 What is the cause of this large  difference within  the industry sector? Coal combustion plays a large role in CO emissions from this sector in the REAS estimate and 98.6% of the combustible industrial emissions are due to coal in 2008. The comparison of fuel use statistics among REAS, EDGAR, and GAINS for 2000 (Fig. 4) and net emission factors per sector among REAS, EDGAR, GAINS, and MEIC (Fig. 5) are useful in understanding the reason behind the dif-

190 ferences. Coal use in industry between REAS and EDGAR shows similar values but there is a large difference in emission factors for industrial CO between REAS (2.2 ton CO/TJ) and EDGAR (1.1 ton CO/TJ). Because emission factors  are related to each technology type, penetration of the technology, uncontrolled emission factor and the emission reduction efficiency of each technology type, these factors all contribute to  differences. Obtaining estimates for CO is particularly

195  difficult because of  many technology types that exist for emissions reduction. For the transport sector, estimated emissions by EDGAR are still lower than those of REAS (Fig. 3), even with its higher fuel use and emission factor, most likely because the modeling of superemitters has been omitted in EDGAR.

200

The smallest CO source sector is power and it has the smallest difference among the inventories.  Power emissions only contribute to 1.9, 3.1, 1.1, and 0.8% of the national emissions in REAS, EDGAR, MEIC, and ZHAO, respectively, in 2008 for the former three and in 2007 for ZHAO. GAINS estimates 1.0% of its national emissions comes from power in 2005. REAS estimates a 2.3

205 Tg (159%) increase in CO emissions from the power sector between 2000 and 2008, while EDGAR only estimates a 0.43 Tg (15%) increase in the same time period. At the national level, the   difference in CO emissions from the power sector between REAS and EDGAR decreased from 50% to 13% between the same period (2000-2008).

The  difference for $PM_{10}$ between REAS and EDGAR is also not insignificant and

210 ranges between 2.7-7.8 Tg yr$^{-1}$ (25 and 59%) over time (Fig. 2). Similar to CO, REAS estimates the highest and EDGAR estimates the lowest national $PM_{10}$ emissions. As shown in Fig. 3, the   major differences arise mainly from the industry sector, where EDGAR emissions show significantly lower estimates compared to those of REAS and  all the others.  Opposite is the case for power sector emissions and

215 EDGAR emissions are double those of REAS and others.    For $PM_{10}$, EDGAR estimates lower fuel use for coal and oil in industry than REAS and higher fuel use for coal and gas in power sector than REAS (Fig. 4). The net emission factor for $PM_{10}$ in industry is

220 also lower for EDGAR than REAS and the opposite is the case for power (Fig. 5). EDGAR thus estimates lower emissions for industry, while estimating higher emissions than REAS for the power sector (Fig. 3). The large  difference in industrial $PM_{10}$ emissions  may also be due to differences in  removal efficiency of a certain technology embedded in emission calculations among inventories.

225

230     The power emissions for $NO_x$ dominate the national total  for REAS, EDGAR, and Zhang et al. (2009) (Fig. 3). 10.9 Tg yr$^{-1}$  (46%) and 10.2 Tg yr$^{-1}$  (51%) of the national $NO_x$ emissions are estimated to come from the power sector in REAS and EDGAR, respectively, in  2008. 9.2 Tg yr$^{-1}$ (44% are estimated to come from the power sector in 2006 in the INTEX-B inventory by (Zhang et al., 2009).) Streets et al. (2003) estimated power

235 to be the dominant source sector,  contributing 4.4 Tg yr$^{-1}$ (39% of $NO_x$ emissions) in 2000, followed by 2.8 Tg yr$^{-1}$ each (equal 25% contribution) from industry and transport. The national emissions inventories, however, do not show  power sector emissions dominating for $NO_x$. For MEIC, industrial emissions are estimated to be slightly higher than those from the power sector. For ZHAO, the two sources are similar in magnitude. 33% (36%) and 35%

240 (35%) of the total emissions equalling 8.6 Tg yr$^{-1}$ (9.4 Tg yr$^{-1}$) and 8.3 Tg yr$^{-1}$ (7.9 Tg yr$^{-1}$) are estimated to come from the power (industry) sector in these two national inventories MEIC in 2008 and ZHAO in 2007, respectively. One of the possible reasons for this

245    is due to the difference in the net emission factors among emission inventories (Fig. 5). MEIC estimates much higher emission factors for $NO_x$ emissions from the industry sector than from power, unlike other inventories that estimate the opposite (REAS and

250 GAINS) or fairly close to each other (EDGAR).

    The  differences for the other species are  smaller, although it is clear that Lamarque et al. (2010) estimates much lower emissions for both $NO_x$ and $SO_2$, compared to others (Fig. 2). The range of the absolute difference between REAS and EDGAR for $CO_2$ and $SO_2$  are  4.25 - 553 Tg yr$^{-1}$ and 0.75 - 7.9 Gg yr$^{-1}$,

255 respectively, between 2000 and 2008. MEIC and ZHAO emissions estimates fall between the REAS and EDGAR estimates most of the time, although they are  closer to the REAS estimates, which are higher than those of EDGAR, for most species. GAINS estimates sometimes do not fall between the REAS and EDGAR estimates but the  differences are still  small. The timing of the $SO_2$ emissions reduction in 2007 in REAS coincides with what is reported in Zhang et al. (2009),

260 Klimont et al. (2009), and Lu et al. (2011).

**3.2 Regional  Level Comparisons**

When we compare emissions in the seven regions within China (East, North, Northeast, Central, Southwest, Northwest, and South, as shown in Fig. 1), we find larger differences than at the national level for almost all species (Figs. 6 - 9). We compare in detail the differences among emissions inventories for each species per region and for each source sector below.

**3.2.1 Carbon monoxide, CO**

Atmospheric CO is mainly a result of incomplete combustion of fossil fuels and biofuelsxposure to ambient CO is harmful to human health (Aronow and Isbell, 1973; Stern et al., 1988; Allred et al., 1989; Morris et al., 1995) CO emissions are also important precursors to the formation of tropospheric $O_3$, which also has harmful human health impacts, including increased asthma exacerbations, decreased pulmonary function, and increased mortality (Schwartz et al., 1994; Mudway and Kelly, 2000; Levy et al., 2005). Because of the existence of diverse emissions sources with various emissions control technologies in China, it has been a challenge to estimate CO emissions accurately, using a bottom-up methodology with emission factors and activity levels (Streets et al., 2006). This explains why we see the largest  difference in CO emissions estimates at the national level  compared in Fig. 2 to all other species.

Fig. 6 shows the  seven regional CO emissions estimates from each source sector. For CO emissions, industry is the only source sector that shows a steep increase over time in all regions for REAS and EDGAR estimates, especially between 2002 and 2008. GAINS also shows an increase between 2000 and 2005.  Due to  the rapid increase in its emissions, by 2008, industry is the largest source sector for CO in the two largest source regions - East and North - regardless of which inventor. REAS CO emissions estimates are consistently higher than those of EDGAR across all regions except for the Northeast for industry emissions, and MEIC, ZHAO, and GAINS CO emissions estimates for this sector generally fall between the estimates of REAS and EDGAR. The two regions where this does not apply are Central and Northwest, and their industrial CO emissions estimates by MEIC, ZHAO, and GAINS are higher than the estimates by the other two emissions inventories. Analysis at the source sector level reveals that the majority of the differences in CO emissions among the inventories stem from the industry sector and that they are, in many regions, increasing over time.

The second largest CO source is the residential sector and the estimates by the national inventories MEIC and ZHAO are always higher in all regions than the regional inventory REAS and the global inventory EDGAR estimates. GAINS estimates the residential sector to be the largest source sector and  their estimates are also usually higher than REAS and EDGAR in almost all regions, except in

the Southwest and the South in 2005, where the REAS and GAINS estimates are close to each other. EDGAR estimates for residential sector emissions  are the lowest among the inventories analyzed here, because it does not include provincial but rather uses the national statistics-based IEA estimates for coal use in residential sector, leading to lower activity level (Fig. 4). On the other hand, GAINS emissions for this sector are the highest because it is unique in considering factors which are technology specific, rather than  one factor per fuel for the whole residential sector . For example, there are significant differences in emissions for different types of stoves and boilers in the residential sector and these technology-specific data are incorporated into the GAINS model.

The third largest CO source and the source sector with the second largest  difference after industry is transport,  contributing 45.6% (34.4%) of the total difference in 2000 (2008) . Emissions from North and East regions contribute to these large  differences. Both REAS and EDGAR emissions inventories show  decrease at the national level, although at the regional scale, the  change is variable, ranging from -0.59 Tg (-1.5 Tg)  for EDGAR  to -1.8 Tg (1.4 Tg) for REAS  between 2000 and 2008 in the North (East). This  difference might be due to a couple of reasons. First, emission factors and reduction measures assumed can be different. For example, EDGAR may be estimating much larger emissions reduction in newer vehicles with more stringent emission standards. Second, the number of vehicles assumed in different vehicle types  is different among the inventories (Fig. 10), even if the total number may be similar. For REAS, the number of vehicles of each type (passenger cars, buses, light and heavy duty trucks, and motorcycles) in 2000 was taken from Borken et al. (2008) and extrapolated to 2008, using trends from National Bureau of Statistics (2001-2009) (Kurokawa et al., 2013). Emission factors due to control strategies and policies in REAS stem from estimates in Borken et al. (2008) and Wu et al. (2011), as explained in Saikawa et al. (2011). For EDGAR, the fleet distribution is based on the international statistics from the International Road Federation (IRF, 1990, 2005, 2007) which were analyzed in the framework of the EU 'Quantify' project (Borken et al., 2008). Zhang et al. (2009) estimated an 11% decrease in CO from the transport sector between 2001 and 2006 due to emissions control technologies, despite the doubling of the number of vehicles in the same period. We will analyze the transport emissions in more detail in Section 3.3 as we have some more disaggregated data for this sector available for comparison.

 At the regional level,  the ranking of source sectors does not always hold and also changes over time. For Northwest, emissions from the residential sector are estimated to be the largest in all years  in all inventories. In Southwest, REAS estimates slightly higher industrial emissions (6.6 Tg yr$^{-1}$ in 2000 and 12.4 Tg yr$^{-1}$ in 2008) than

residential emissions (6.3 Tg yr$^{-1}$ in 2000 and 9.9 Tg yr$^{-1}$ in 2008) but EDGAR estimates higher transport emissions (2.5 Tg yr$^{-1}$) than industrial (2.0 Tg yr$^{-1}$) in 2000. Similarly, in the South, REAS estimates industry to be the largest source sector (6.4 Tg yr$^{-1}$) followed by residential (5.3 Tg yr$^{-1}$) and transport (4.5 Tg yr$^{-1}$) in 2008, whereas EDGAR estimates residential to be the largest (3.7 Tg yr$^{-1}$), followed by industry  in a close second (3.4 Tg yr$^{-1}$) and transport (0.73 Tg yr$^{-1}$) with much lower emissions than the other two in the  same year. This clearly illustrates the importance of constraining emissions at the disaggregated levels.

The East, encompassing the Pearl-River-Delta and the industrial coast, is the largest source region of CO.  In 2008, 32, 27, and 26% of the national total CO emissions from REAS, EDGAR, and MEIC estimates, respectively were emitted from this region. Similarly, ZHAO (GAINS) estimates 30% (29%) of the national total CO emissions is from the East in 2007 (2005). CO emissions from the industry sector in the East, in particular, show  large differences, and the absolute difference more than doubles from 2000 to 2008. In 2008, there is a 22.4 Tg yr$^{-1}$ difference in CO emissions  within  the industry sector  between REAS and EDGAR, which constitutes a 64%  difference between the two emissions estimates in the East in that year. This  difference makes up 25% of the difference between the two national total CO emissions estimates. The difference between the REAS and EDGAR emissions estimates for the transport sector for this region is also increasing and is 10.1 Tg yr$^{-1}$ in 2008, equivalent to 29% of the regional total CO difference and 11% of the national CO difference. One thing to note about this region is that EDGAR CO estimates for the transport sector are decreasing over time, whereas those of REAS indicate the opposite.

The North is the second largest source region of CO, and it  contributes 21, 14, and 21% of the national total CO emissions for REAS, EDGAR, and MEIC estimates, respectively, in 2008. ZHAO (GAINS) estimates 18.5% (18.1%) of the national total CO emissions come from this region in 2007 (2005). Combined with the East emissions, the two regions contribute 53, 42, 47, 48, and 47% of the emissions in REAS, EDGAR, MEIC, ZHAO, and GAINS, respectively, in 2008 for the former three, 2007 for ZHAO, and 2005 for GAINS. The pattern shown for East and North, the more developed regions in China, is similar, and the only difference is that EDGAR estimates larger residential emissions compared to transport emissions in the East, whereas the opposite is the case for the North in the early 2000s.

**3.2.2  Sulfur dioxide, SO$_2$**

SO$_2$ leads to acid rain through sulfuric acid deposition, destroying buildings by corroding metals and deteriorating paint and stone. Furthermore, it harms aquatic and terrestrial ecosystems. SO$_2$ is also a precursor of sulfate aerosols that scatter radiation, leading to direct cooling of the atmosphere. Sulfate aerosols also act as condensation nuclei, making clouds more reflective and prolonging the

lifetime of clouds, enhancing the cooling impact (Haywood and Boucher, 2000; Ramanathan et al., 2001).

Fig. 7 shows the  seven regional $SO_2$ emissions estimates for each source sector.

375 For $SO_2$ emissions, the power sector is the largest source sector in most years for both REAS and EDGAR, and 38 - 54% (52 - 61%) of national total $SO_2$ emissions are from the power sector in REAS (EDGAR) between 2000 and 2008. Contrary to CO emissions, we find a large divergence between REAS and EDGAR power sector emissions estimates during 2000 - 2008 across all regions. While EDGAR $SO_2$ power emissions estimates continue to increase over time, those of REAS peak in that

380 time range, although the specific year is not uniform across the regions. Up to the peak in the REAS estimates, REAS and EDGAR follow similar trajectories. However, REAS $SO_2$ emissions in t Central and the Northwest start to  decrease in 2004, in 2005 in the South, East, and North , and in 2006 in the Northeast and the Southwest .

The large  difference in $SO_2$ emissions from the power sector between REAS and

385 EDGAR is due to the difference in the assumed timing of the installation of FGD in coal-fired power plants. Newly designed policy incentives and an increase in policy inspection have led to an increase in the installation of FGD in China and the percentage of plants with FGD increased from 10% to 71% between 2006 and 2009 (Xu, 2011). The number of power plants is listed in Table 3. While EDGAR assumed a delayed penetration of FGD (1%), electrostatic precipitators (6%) and flue-gas

390 recirculation (4%) leaving 90% of power plants still fully-uncontrolled in 2008, REAS estimated a more optimistic installation scenario, especially for large power plants and referred to Lu et al. (2011) in deciding implementation rates of FGD to power plants in China. For example in 2007, Lu et al. (2011) used the range of 51.4 - 95%, with the mean of 73.2%, based on the Chinese Ministry of Environmental Protection (MEP) official data (2009) reporting of $SO_2$ removal efficiency

395 of FGD and applying the triangular distribution with the ideal removal efficiency of 95% (Zhao et al., 2011). This explains why REAS emissions estimates from the power sector are closer to the  emissions estimates by MEIC, and those by Lu et al. (2011), as seen in Figs. 3 and 7. The largest emissions decrease from the power sector are seen in the East and North regions, where there were 250 and 206 power plants, respectively, reinforcing that this  difference is due to the

400 FGD implementation assumption in power plants.

The second largest source sector for China's $SO_2$ emissions is industry. Nationally, it  contributes 13 (53%), 17 (33%), 17 (53%), 14 (44%), and 9.3 (27%) Tg yr$^{-1}$ of total $SO_2$ emissions in REAS, EDGAR, and MEIC for 2008, ZHAO for 2007, and GAINS for 2005, respectively. In some regions, such as the Northeast, there is very little  difference among inventories

405 . On the other hand, we see a much larger difference in the Southwest. While EDGAR estimates industry to be the second largest source sector in this region, constituting 31 - 37% of regional emissions, all other emissions inventories estimate industry to be the largest source sector in the region, constituting 46 - 60% of the regional total. Similar to its estimates for

CO emissions, REAS tends to estimate higher emissions from the industry sector in most of the regions.

$SO_2$ emissions discrepancies in the two other sectors remain relatively small and constant across all regions, with the residential sector emissions in the Southwest as the only exception. The  residential sector emissions difference in the Southwest between EDGAR and REAS estimates  have decreased from  354 Gg yr$^{-1}$ in 2000 to  215 Gg yr$^{-1}$ in 2008.

**3.2.3 Nitrogen oxides, $NO_x$**

$NO_x$ plays an important role in the formation of tropospheric $O_3$ and nitrate aerosols. The $NO_x$ emissions trend in Asia, and especially in China, has been an important topic, due to the rapid changes that have been observed in the past two decades (Richter et al., 2005; Gu et al., 2014). Fig. 8 shows the  seven regional $NO_x$ emissions estimated for each source sector. The  difference in this sector is the largest in the East and the Northeast regions, both with 470 Gg yr$^{-1}$ in 2008.

The large  differences among the emissions inventories stem from the transport sector in the East, North, South, and Northwest . For the transport sector, the East has an increasing  difference over time, changing from  0.40 Tg in 2000 to  1.3 Tg in 2008. While transport contributes 27 - 30% of the regional total emissions for REAS in the East, it only  contributes 15 - 19% for EDGAR. MEIC estimates the transport sector in the East to  contribute 25% of the regional total $NO_x$ emissions. In the North, South, and Northwest,  the  difference in the transport sector emissions among the inventories can also be as high as 450, 355, and 326 Gg yr$^{-1}$, respectively. The key reasons why the differences are large and they are growing are two-folds. First, as we explain later in Section 4,  the differences in the allocation of fuel (gasoline and diesel) and tces in vehicle categories play a role. Second,  the pace of the implementation of measures assumed among different inventories is different.

Little to no emissions control technologies for $NO_x$ has been developed and promoted in China for the power and industrial combustion sectors and this is the main reason why we see a large  increase for $NO_x$ emissions. China only used low-$NO_x$ combustion technology and started to install selective reduction methods after 2005 (Zhao et al., 2013a). The only other $NO_x$ mitigation strategy for China was emissions standards for reducing tail pipe emissions from vehicles (Zhao et al., 2013a). For example, there is no national $NO_x$ emissions standard for coal-fired industrial boilers, as opposed to the vehicle emission standards that have been tightened over the years.

**3.2.4 Coarse particulate matter, $PM_{10}$**

China's $PM_{10}$ emissions have been increasing rapidly and they  contribute approximately  21.6 (15.2) Tg yr$^{-1}$ of  38.3 (39.3) Tg yr$^{-1}$ total $PM_{10}$ emissions from 22 Asian countries, including Afghanistan, Bangladesh, Bhutan, Nepal, Sri Lanka, India, Maldives, Pakistan, South Korea, North Korea, China, Japan, Singapore, Taiwan, Laos, Cambodia, Brunei, Myanmar, Philippines, Thailand, Vietnam, and Indonesia, in the REAS (EDGAR) estimate. These differences between REAS and EDGAR estimates indicate the large differences in China, as well as in other parts of Asia. Here, we only discuss primary emissions of $PM_{10}$, emitted directly from anthropogenic sources.

Fig. 9 shows the  seven regional $PM_{10}$ emissions estimates for each source sector. The largest source sector, as well as the largest emissions  difference, stems from the industry sector. Industrial emissions contribute 64, 19, and 78% of the total $PM_{10}$ emissions in REAS, EDGAR, and MEIC for 2008, respectively, and 65% (50%) for ZHAO (GAINS) for 2007 (2005).  As illustrated in the low industrial contribution in 2008 in EDGAR  its industrial emissions  increased by 1.3 Tg from 2000 to 2008,  while those of REAS  increased by 5.8 Tg in the same period. The reason for this large increase in industrial $PM_{10}$ emissions is due to the fast growth of industry and limited stringency of air quality legislation and its poor enforcement (Zhao et al., 2013a). In addition, uncertainty accounting for fugitive emissions due to leaks or other unintentional releases adds to the  difference among the inventories. For industrial $PM_{10}$ emissions, REAS estimates are always consistently higher than those of EDGAR in all regions, and the difference between the two inventories is four to five-fold, constituting 61 - 74% of the total differences.

We see relatively little change in differences among the inventories between 2000 and 2008 for transport and residential sectors. It is also important to point out that the spatial distribution of emissions in some of the inventories, especially the global ones, are often more static than the national ones due to the limited local information, although this static nature over time of the global inventories is not only for $PM_{10}$ but also applies to other species as well. There are, however, some interesting sector-dependent differences. First, GAINS estimates higher residential emissions than REAS and EDGAR in all regions in both 2000 and 2005 except in the South in 2005. Second, REAS estimates are not always higher than those of EDGAR for the residential sector emissions. In the Northeast, REAS $PM_{10}$ emissions estimates are higher than those of EDGAR. For the Southwest and the North, REAS emissions estimates are higher than EDGAR estimates only for the period 2002 - 2005.

**480  4    Road Transportation Sector Comparison**

Rapid growth in the number of vehicles has created a significant air quality challenge in China. Many have researched the importance of on-road transportation emissions on Beijing's (Hao et al., 2001; Westerdahl et al., 2009) and China's air quality (Fu et al., 2001; Walsh, 2007; Saikawa et al., 2011). We found significant discrepancies differences in CO, and $NO_x$, $PM_{10}$, and $SO_2$ emissions in the

485    transportation sector and here we analyzed the differences for CO and $NO_x$ these emissions in more depthbecause we were able to by focusing on these to both on-road and off-road transport emissions. Here, we first explain the discrepancies we find for each of the species. compare the contribution of different vehicle categories to the total vehicles in REAS, EDGAR, and GAINS. Then, we compare on-road and off-road emissions estimates of CO, $NO_x$, $SO_2$, and $PM_{10}$ at the national level, as well

490    as for each region.

Comparing the contribution of various gasoline (Fig. 10a) vehicles among the three inventories, EDGAR is very different from the other two. The similar comparison for diesel vehicles (Fig. 10b) reveals even a larger difference. As stated earlier for the industrial sector, it is likely that emission factors and/or the technology levels estimated within each of the vehicle types are causing discrep-

495    ancies the differences. EDGAR emission factors specifically for on-road vehicles is not available but comparing the net transport-sector emission factors between EDGAR and GAINS (Fig. 5), GAINS has 5.6 times higher value per unit of fuel than EDGAR. The lack of modeling superemitters in EDGAR is also contributing significantly to the discrepancies differences. It is also possible that something more fundamental, such as the definition of vehicle types, is possibly causing the differ-

500    ences. It is also important to keep in mind that something more fundamental, such as the definition of vehicle types, is possibly causing the discrepancies.

In the following section, we compare national on-road and off-road transport emissions first among REAS, EDGAR, ZHAO, and GAINS, and then in the seven regions within China (East, North, Northeast, Central, Southwest, Northwest, and South, as shown in Fig. 1), for REAS, EDGAR,

505    and ZHAO. We compare in detail the differences among emissions inventories for each species per region and for each source sector below.

**4.1    Carbon monoxide, CO**

Fig. 11 shows the national and seven regional CO transportation emissions estimated in REAS, EDGAR, ZHAO, and GAINS (national estimate only), separated into on-road and off-road emis-

510    sions. and it The figure clearly shows clearly that the discrepancy difference in this sector stems from on-road emissions. 99% of the difference between REAS and EDGAR CO transportation emissions are from on-road at the national level, and in the East, we see up to a difference of 99.4% at the regional level. Indeed, at the national and all regional levels, there is more than an order of magnitude of difference in emissions between REAS and EDGAR on-road emissions. ZHAO on-road emis-

515 sions estimates are always in between REAS and EDGAR estimates and ZHAO off-road estimates
are always higher than both REAS and EDGAR.

**4.2 Nitrogen Oxides, $NO_x$**

Fig. 12 shows the national and seven regional $NO_x$ transport emissions estimated in REAS,
EDGAR, and ZHAO, separated into on-road and off-road emissions. Contrary to the CO emis-
520 sions, there are many regional differences in these emissions estimates. At the national level, REAS
(ZHAO) estimates 42-56% (49%) higher for on-road emissions compared to EDGAR. Off-road
emissions are much more constrained among the three emissions inventories and REAS and EDGAR
give similar estimates between 2005 and 2007.

The East is estimated to  contribute 28-38, 6.3-6.8, and 37% of the total transport emis-
525 sions in REAS, EDGAR, and ZHAO, respectively. REAS (ZHAO) emissions estimates are 5.6-7.4
(6.2) times larger than EDGAR on-road emissions, and 2.6-9.5 (6.7) times larger than off-road emis-
sions. For $NO_x$ emissions, although on-road emissions are still larger in most of the regions, off-road
emissions are also important and are mostly increasing in both REAS and EDGAR. For the East,
REAS estimates an increase  from 307 Gg yr$^{-1}$ in 2000 to 1100 Gg yr$^{-1}$ in 2008 in off-road
530 emissions. For the Northwest, EDGAR estimates larger emissions from off-
road compared to on-road for $NO_x$, which we do not see in either REAS or ZHAO. REAS estimates
a higher growth rate for off-road emissions and their emissions estimates increase  from
28.4 Gg yr$^{-1}$ in 2000 to 75.1 Gg yr$^{-1}$ in 2008, while EDGAR off-road emissions estimates only
increase  from 98.5 Gg yr$^{-1}$ to 110 Gg yr$^{-1}$ over the same time period. The large emissions
535 differences in the region are most likely due to much greater railway emissions by coal and diesel
locomotives assumed in the EDGAR inventory, compared to REAS.

**4.3 Coarse Particulate Matter $PM_{10}$ and Sulfur dioxide $SO_2$**

Fig. 13 shows the national $PM_{10}$ and $SO_2$ on-road and off-road emissions estimated in REAS,
EDGAR, and GAINS. $PM_{10}$ shows a good agreement for on-road emissions between REAS and
540 GAINS, although EDGAR on-road is much lower. The low emissions estimates for EDGAR for
$PM_{10}$ is most likely due to the lack of superemitters in EDGAR, since those are the primary emit-
ters. On-road emissions for $SO_2$ also shows a good agreement, especially between EDGAR and
GAINS, although REAS values show an increase in the late 2000s that we do not find in the other
two inventories. $SO_2$ is calculated differently than for the other species in REAS, based on gaso-
545 line/diesel consumption instead of vehicle category. This might also be the reason for the difference
among the inventories.

Off-road emissions are in especially good agreement for $PM_{10}$ among the three inventories. How-
ever, they diverge quite significantly for $SO_2$ emissions. GAINS, in particular, has low emissions
estimates for off-road $SO_2$ emissions, although it estimates high emissions for CO and $PM_{10}$. Based

550 on Fig. 5, it is most likely due to the high emission factors GAINS have for these off-road vehicles in the transport sector.

**5 Impacts on air quality**

**5.1 Model description**

To assess how these differences in emissions inputs affect air quality simulation results, we used the
555 Weather Research and Forecasting model coupled with Chemistry (WRF-Chem) version 3.5 (Grell et al., 2005). The model domain covers much of the Asian region, with a horizontal resolution of 20 × 20 km  with 31 vertical levels and China at its center (Fig. 15). The initial and lateral chemical boundary conditions are taken from a present-day simulation of the NOAA Geophysical Fluid Dynamics Laboratory (GFDL) global chemistry-climate model AM3 (Naik et al., 2013), driven by the
560 global gridded emissions from the inventory of Lamarque et al. (2010). The meteorological data are obtained from the National Center for Environmental Prediction (NCEP) Global Forecast System final gridded analysis datasets. We used Carbon-Bond Mechanism version Z (CBMZ) (Zaveri and Peters, 1999) for gas-phase chemistry and the Model for Simulating Aerosol Interactions and Chemistry (MOSAIC) (Zaveri et al., 2008) for aerosol chemistry. The rest of the model setup (aerosol dry
565 deposition, wet deposition, photolysis, radiation, and microphysics) is the same as applied in our previous study (Zhong et al., 2015).

We chose the three emissions inventories that provided gridded emissions and are targeted at different scales:  EDGAR at global, REAS at regional, and MEIC at national. In addition, EDGAR estimates the lowest emissions for most species, whereas REAS estimates the highest and
570 thus provides a range of air quality simulations as a result of varying emissions.  We then  performed model simulations for January and July for 2008, using each of these inventories. Because MEIC only covers China, we applied REAS emissions outside of China for the simulation with MEIC. For biomass burning emissions, we used the Fire INventory from NCAR (FINN) (Wiedinmyer et al., 2011) and for biogenic emissions, we
575 used the Model of Emissions of Gases and Aerosols from Nature (MEGAN) interactively within WRF-Chem (Guenther et al., 2012). For aircraft emissions, we used emissions developed for the Hemispheric Transport of Air Pollution (HTAP) for the year 2008 (Janssens-Maenhout et al., 2015). In order to focus on differences in air quality due to differing anthropogenic emissions estimates of gaseous pollutants and PM, we did not include dust simulation in this study.
580  However, sea salt is calculated online (Gong, 2003). Before the beginning of each monthly simulation, the model was spun-up for ten days to ventilate the regional domain. The model simulation including dust has been validated with existing measurements for the year 2007 in (Zhong et al., 2015) and here we focus on differences in air quality simulation due to differing gridded anthropogenic emissions inputs.

 **5.2  Simulated results and discussion**

Fig. 14a illustrates the spatial distribution of January emissions for CO, NO$_x$, SO$_2$, PM$_{10}$, and NMVOC that we used as inputs for the WRF-Chem simulations. As mentioned earlier, CO and PM$_{10}$ show high variations and the emissions are especially concentrated in the eastern part of China. Although the difference in national SO$_2$  emissions was not as large as those of the other two species, Fig. 14a clearly illustrates that REAS estimates much larger emissions compared to the other two inventories.

Fig.15a compares the simulated monthly mean PM$_{10}$ concentrations, as well as that of CO, NO$_2$, SO$_2$, and O$_3$ mixing ratios in January 2008, using the three inventory estimates as emissions inputs. These differences in simulated concentrations or mixing ratios of pollutants are solely due to the emissions used as model inputs. Overall, the simulated monthly means show similar spatial distributions. All three simulations show high levels of CO, NO$_2$, SO$_2$, and PM$_{10}$ in the Beijing-Tianjin-Hebei area in the North, Shanxi province in the North, and Sichuan basin in the Southwest. In contrast, the mixing ratios of O$_3$ are relatively low over the same regions. Despite the similar spatial distributions,  concentrations of the simulated monthly means differ substantially.

For CO, both simulations using REAS and MEIC result in higher mixing ratios than when using EDGAR . We quantified the regional monthly mean of each simulation by averaging all grid cells in each region, as illustrated in Table 4. The REAS and MEIC regional monthly means are  270 - 470 ppbv (169 - 194 ppbv) higher in the polluted area in the Central  (the East) region, than the EDGAR simulation.  For NO$_2$, the largest differences in regional monthly mean occur between simulations using EDGAR and MEIC emissions, mainly in the Central ( 8.1 ppbv), followed by the  East ( 7.2 ppbv) and the  Northeast ( 3.3 ppbv). These regions are where the differences in emissions are the largest as well. For SO$_2$, both simulations using REAS and MEIC show differences in monthly mean less than 30% in most regions compared to those with EDGAR emissions, except in the Southwest, where REAS and MEIC estimates are  1.5 and 1.7 ppbv higher, respectively, than EDGAR estimates.

For PM$_{10}$, EDGAR simulation is 20 - 60 $\mu$g $m^{-3}$ lower than the other two in most regions. For example, MEIC simulation estimates 15 $\mu$g $m^{-3}$ (103%) higher monthly mean in the Northeast and 20 $\mu$g $m^{-3}$ (85%) higher in the Southwest than EDGAR. REAS simulation estimates more than 55% higher monthly mean PM$_{10}$ concentrations than EDGAR in most regions, with the highest difference (76%) occurring in the Northeast. The largest absolute difference of 67 $\mu$g $m^{-3}$ in a regional monthly mean between MEIC and EDGAR simulations is found in  Central . Based on the observations from nine stations in Wuhan within the Central region, the monthly mean PM$_{10}$ concentrations in January were 130 $\mu$g $m^{-3}$ (Feng et al., 2011) and this is closer to the simulated values using the MEIC (REAS) emissions inventory of 47.4 (50.6) $\mu$g $m^{-3}$, compared to

the value using the EDGAR emissions inventory of 32.3 $\mu$g $m^{-3}$, although the model simulations are largely underestimated.

For $O_3$, simulations using REAS and EDGAR inputs show only a slight difference in monthly mean of 1-5 ppbv in January. However, $O_3$ mixing ratios using MEIC emissions are much lower than those using EDGAR emissions in the Central (31%) and the East (25%). MEIC's low anthropogenic VOC emissions in combination with high $NO_x$ emissions in these regions (see 14a) bring much higher $NO_x$ titration and produce a VOC-limited environment. It is well, as illustrated in Figure 16a. For these two regions, despite the REAS and MEIC having similar $NO_x$ emissions, their VOC emissions differ by more than 10 times. EDGAR emissions are the lowest for $NO_x$ for both the Central and the East but their estimates are the largest for VOCs in the Central and the second largest in the East among the three inventories. In both cases, simulations using EDGAR inventory lead to the largest $O_3$ mixing ratios, due to the limited titration of $NO_x$ during the night time. The $NO_x$ mixing ratio in these two regions estimated in EDGAR is much lower compared to that in REAS and MEIC, as seen in Fig. 15a. This result illustrates the importance of VOC emissions estimates, in addition to $NO_x$ and other species that we have analyzed in this paper. Constraining these $NO_x$ and VOC emissions in the two East and Central regions will be essential in understanding the way to mitigate $O_3$ pollution for the future.

We also analyzed the differences of three simulations in July 2008 (Fig. 15b). We find a difference of more than 50% for CO, $NO_2$, $SO_2$, and $PM_{10}$ in one or more regions , while a difference is less than 20% for $O_3$ in every region. The Central and the East again showed the largest differences, as found in January. There was a 34 $\mu$g $m^{-3}$ difference in $PM_{10}$ in Central China between REAS and EDGAR and a 129 ppbv difference in the East for CO between REAS and MEIC. Again, Wuhan mean for July of 70 $\mu$g $m^{-3}$ of $PM_{10}$ was better captured by MEIC (REAS) of 52.0 (53.5) $\mu$g $m^{-3}$, compared to that by EDGAR of 36.0 $\mu$g $m^{-3}$. The difference we find for $O_3$ in East, North, and Central are also important, due to the high mixing ratio estimated in REAS is close to the 8-hr WHO guideline of 100$\mu$g m$^{-3}$. From Fig. 16b, it is clear that the difference of $O_3$ mixing ratio in these three regions is due almost solely to the VOC emissions between REAS and MEIC. More detailed comparisons are illustrated in Table 4. These differences in simulated concentrations or mixing ratios of pollutants are solely due to the emissions used as model inputs. Not surprisingly, the results demonstrate that the choice of emissions inventories has a large influence on air quality simulation results and reinforce the need for better constraints on emissions inputs.

**6 Conclusions**

In this study, we compared five emissions inventories of anthropogenic $CO_2$ and air pollutant emissions in China at national and regional levels from four source sectors. The REAS and EDGAR inventories have been developed and maintained for years and have been extensively used for air quality modeling over the Asian continent, while the two  national emissions inventories (MEIC and ZHAO) were recently developed, and  few air quality modeling studies have been published using the data from these inventories at this time. GAINS has its roots in the RAINS-Asia model dating back to early 90's project covering primarily $SO_2$ and later on developed to include more pollutants. The GAINS dataset used here originates from a global project and has been used in several air quality and climate modeling exercises. This analysis reveals large  differences in emissions estimates among the existing inventories. Furthermore, analysis of regional and sector specific emissions, as opposed to total national emissions, reveals  differences in emissions from certain sectors that would not have been noticed by only analyzing the national total emissions.

We find that there is  a significant need to better constrain emissions at the source sector and regional levels. Transparency in what inputs are used to create different emissions inventories  is critical for a more thorough comparison. CO emissions differ the most, and those from the transport sector, especially the on-road transport emissions, need to be better constrained. Industrial emissions also tend to have a large  difference among inventories and $SO_2$ emissions from the power sector also need to be assessed, especially for recent years. The East and the North are the two largest emitting regions and more efforts are needed to understand emissions from these areas.

Emissions inputs have a large impact on air quality simulation results in China nationally, and more prominently within the regions. Different emissions inputs lead to 67 $\mu g\ m^{-3}$ (34 $\mu g\ m^{-3}$) monthly mean difference in $PM_{10}$ concentrations in Central China in January (July). Similarly, we found 470 ppbv difference in January in Central and 129 ppbv difference in July in the East for CO. We also found that  all the three inventory emissions estimates create a VOC-limited environment in the Central and the East that produces much lower $O_3$ mixing ratio estimates, compared to the simulations using REAS and EDGAR estimates in January. The  difference in emissions inputs leads to 15 ppbv difference in $O_3$ in Central China in January. In July, we find 8.5 ppbv difference in North, where REAS simulations lead to a monthly-mean of 63 ppbv $O_3$. Our results illustrate that a better understanding of Chinese emissions at more disaggregated levels is essential for finding  effective mitigation measures for reducing national and regional air pollution in China.

*Acknowledgements.* We thank the editor and the two anonymous reviewers for constructive comments that improved the manuscript. We also thank Geoffrey Martin and Raquel Soat for their assistance in this project. The project was partly supported by the National Science Foundation (grant number AGS-1350021). REAS is supported by the Global Environmental Research Fund of the Ministry of the Environment Japan (S-7 and S-12). The NCEP GFS data used for this study are from the Research Data Archive (RDA) which is maintained by the Computational and Information Systems Laboratory (CISL) at the National Center for Atmospheric Research (NCAR). The data is available at http://rda.ucar.edu/datasets/ds083.2/. We would like to acknowledge

high-performance computing support from Yellowstone (ark:/85065/d7wd3xhc) provided by NCAR's Computational and Information Systems Laboratory, sponsored by the National Science Foundation.

**References**

[revised manuscript text omitted]

| | Years | Source Sectors | Species | Horizontal Resolution | Coverage | Reference |
|---|---|---|---|---|---|---|
| REAS | 2000-2008 | power plants, combustible and non-combustible sources in industry, on-road and off-road sources in transportation, residential, agricultural, and other anthropogenic sources | $CO_2$, $SO_2$, CO, $PM_{10}$, $PM_{2.5}$, BC, OC, $NO_x$, $NH_3$, NMVOC, $CH_4$, $N_2O$ | 0.25° x 0.25° | East, Southeast, South & Central Asia. Asian part of Russia | Kurokawa et al., 2013 |
| EDGAR | 1970-2008 | energy, industrial processes, product use, agriculture, large scale biomass burning, and other anthropogenic sources | $CO_2$, $SO_2$, CO, $PM_{10}$, $NO_x$, $NH_3$, NMVOC, $CH_4$, $N_2O$ HFCs, $SF_6$, $NF_3$ | 0.1° x 0.1° | Global | EC-JRC/PBL, 2011 |
| MEIC | 2008, 2010 | power, industry, transportation, residential and agricultural sources | $CO_2$, $SO_2$, CO, $PM_{10}$, $NO_x$, NMVOC | 0.1° x 0.1° | China | www.meicmodel.org |
| ZHAO | 2000-2014 | power, combustible and non-combustible sources in industry, on-road and off-road sources in transportation and residential | $CO_2$, $SO_2$, CO, TSP, $PM_{10}$, $PM_{2.5}$, BC, OC, $NO_x$, Hg | N/A | China | Zhao et al., 2013b Zhao et al., 2015 Cui et al., 2015 Xia et al., 2016 |
| GAINS | 1990-2030 (5-yr increment, projection starting in 2015) | energy, domestic, industrial combustion and processes, road and non-road transportation and agriculture | $CO_2$, $SO_2$, CO, TSP, $PM_{10}$, $PM_{2.5}$, $PM_1$, BC, OC, $NO_x$, $NH_3$, VOC, $CH_4$, $N_2O$, F-gases | 0.5° x 0.5° | Global | Amann et al., 2011 Klimont et al., in review Klimont et al., in preparation |

**Table 2.** Source categorizations

| | EDGAR | REAS | ZHAO | MEIC | GAINS |
|---|---|---|---|---|---|
| Industry | Manufacturing industries and construction
Production of minerals
Production of chemicals
Production of metals
Production of pulp/paper/food/drink
Production of halocarbons and $SF_6$

Refrigeration and air conditioning
Foam blowing
Fire extinguishers
Aerosols
F-gas as solvent
Semicondutor/electronics manufacturing
Electrical equipment
Other F-gas use
Solvent and other product use | **Combustible:** Iron and steel, Chemical and petrochemical, Non-ferrous metal, Non-metallic minerals, Energy, Others

**Non-combustible:** Pig iron, Crude steel, Iron steel others, Aluminum & Alumina, Copper, Zinc, Lead, Cement, Bricks, Lime, Coke ovens, Oil refinery, Other transformation, Sulphuric acid, Others | Industry | Industry | **Combustible:** Iron and steel, Pulp and Paper, Chemical, Non-ferrous metals, Non-metallic minerals, Other

**Processes:** Pig iron, Coke ovens, Agglomeration plants, Steel, Rolling mills, Cast iron, Non-ferrous metals, Cement & Lime, Sulfuric acid, Nitric acid, Aluminium, Aluminium, Glass production, Fertilizer production, Brick manufacturing, Pulp and paper, Refineries, Others |
| Transportation | Domestic aviation
Road transportation
Rail transportation
Domestic navigation
Other transportation | Cars
Buses
Light trucks
Heavy trucks
Motorcycles
Other vehicles
Domestic navigation
Railway & etc. | Light duty vehicles
Rural vehicles
Small gasoline engines
Heavy duty vehicles
Motorcycles
Machines
Inland shipping
Railway | Transportation | Cars
Buses
Light duty vehicles
Heavy duty vehicles
Motorcycles
Mopeds
Domestic navigation
Railway & etc. |
| Power | Fugitive emissions from solid fuels
Fugitive emissions from oil and gas
Public electricity and heat production
Other energy industries
Non-energy use of lubricants/waxes ($CO_2$)
Fossil fuel fires | Power plants | Power | Power | Power plants
Diesel generators
Briquette production
Extraction and distribution of solid fuels
Extraction and distribution
of liquid & gaseous fuels |
| Residential | Residential and other sectors
Waste incineration | Residential and other sectors | Residential and other sectors | Residential and other sectors | Cooking and heating
Kerosene lighting
Waste (trash) burning |

**Table 3.** Number of power plants in each region within China

| Region | Number of coal power plants |
| --- | --- |
| East | 250 |
| North | 206 |
| Central | 86 |
| South | 78 |
| Northeast | 76 |
| Southwest | 66 |
| Northwest | 43 |

Source: Carbon Monitoring for Action

**Table 4.** Regional monthly mean concentrations of MEIC, REAS, and EDGAR and largest differences found within a region in WRF-Chem simulation in 2008

(a) January

| Regions | $PM_{10}$ ($\mu g/m^3$) | | | | $O_3$ (ppbv) | | | | $SO_2$ (ppbv) | | | | $NO_2$ (ppbv) | | | | CO (ppbv) | | | |
|---|---|---|---|---|---|---|---|---|---|---|---|---|---|---|---|---|---|---|---|---|
| | MEIC | REAS | EDGAR | diff | MEIC | REAS | EDGAR | diff | MEIC | REAS | EDGAR | diff | MEIC | REAS | EDGAR | diff | MEIC | REAS | EDGAR | diff |
| Central | 163 | 155 | 96 | 67 | 31 | 41 | 46 | 15 | 26 | 23 | 21 | 5.3 | 15 | 12 | 6.9 | 8.1 | 852 | 632 | 382 | 470 |
| East | 123 | 129 | 82 | 48 | 32 | 39 | 43 | 11 | 15 | 16 | 18 | 3.1 | 15 | 13 | 8.0 | 7.2 | 623 | 598 | 329 | 294 |
| North | 27 | 27 | 19 | 8.5 | 45 | 46 | 47 | 2.0 | 8.1 | 7.1 | 9.3 | 2.2 | 5.6 | 5.1 | 3.9 | 1.7 | 255 | 214 | 147 | 108 |
| Northeast | 29 | 25 | 14 | 15 | 41 | 41 | 46 | 4.5 | 4.2 | 5.1 | 5.9 | 1.7 | 6.6 | 6.5 | 3.4 | 3.3 | 259 | 242 | 165 | 94 |
| Northwest | 19 | 19 | 14 | 4.9 | 55 | 56 | 56 | 1.0 | 3.9 | 3.4 | 4.0 | 0.59 | 1.5 | 1.4 | 1.2 | 0.3 | 166 | 154 | 119 | 47 |
| South | 127 | 128 | 82 | 46 | 40 | 47 | 44 | 6.8 | 6.3 | 7.7 | 8.6 | 2.3 | 4.5 | 4.0 | 3.4 | 1.1 | 534 | 548 | 321 | 228 |
| Southwest | 42 | 37 | 23 | 19 | 59 | 60 | 59 | 1.2 | 5.1 | 4.9 | 3.4 | 1.7 | 1.5 | 1.4 | 1.3 | 0.13 | 284 | 242 | 156 | 128 |

(b) July

| Regions | $PM_{10}$ ($\mu g/m^3$) | | | | $O_3$ (ppbv) | | | | $SO_2$ (ppbv) | | | | $NO_2$ (ppbv) | | | | CO (ppbv) | | | |
|---|---|---|---|---|---|---|---|---|---|---|---|---|---|---|---|---|---|---|---|---|
| | MEIC | REAS | EDGAR | diff | MEIC | REAS | EDGAR | diff | MEIC | REAS | EDGAR | diff | MEIC | REAS | EDGAR | diff | MEIC | REAS | EDGAR | diff |
| Central | 64 | 72 | 37 | 34 | 55 | 62 | 56 | 6.9 | 7.3 | 7.0 | 7.1 | 0.27 | 5.4 | 6.7 | 4.0 | 2.7 | 263 | 300 | 224 | 77 |
| East | 56 | 64 | 36 | 28 | 55 | 63 | 55 | 8.1 | 6.9 | 8.4 | 9.4 | 2.5 | 7.8 | 8.7 | 5.0 | 3.7 | 247 | 321 | 192 | 129 |
| North | 39 | 33 | 21 | 13 | 58 | 63 | 54 | 8.5 | 4.2 | 4.5 | 5.6 | 1.3 | 2.6 | 3.0 | 2.0 | 1.0 | 178 | 212 | 130 | 82 |
| Northeast | 39 | 33 | 21 | 12 | 51 | 55 | 47 | 8.2 | 1.5 | 2.2 | 3.3 | 1.8 | 2.7 | 3.1 | 1.9 | 1.2 | 172 | 199 | 153 | 46 |
| Northwest | 8.4 | 8.8 | 6.3 | 2.5 | 55 | 58 | 53 | 4.8 | 1.4 | 1.3 | 1.6 | 0.22 | 0.57 | 0.78 | 0.61 | 0.21 | 94 | 95 | 90 | 5.3 |
| South | 19 | 21 | 17 | 3.8 | 39 | 44 | 40 | 5.0 | 2.4 | 4.3 | 5.2 | 2.9 | 2.9 | 4.0 | 3.1 | 1.1 | 170 | 185 | 156 | 29 |
| Southwest | 11 | 12 | 7.9 | 4.1 | 50 | 53 | 50 | 3.6 | 1.8 | 2.3 | 1.5 | 0.75 | 0.93 | 1.3 | 1.1 | 0.34 | 116 | 125 | 104 | 21 |